# Motivations for the Use of IoT Solutions by Company Managers in the Digital Age: A Romanian Case

**Mirela Cătălina Tűrkeș** [1,*] , **Sorinel Căpușneanu** [2] , **Dan Ioan Topor** [3], **Adela Ioana Staraș** [4], **Mihaela Ștefan Hint** [3] **and Laurentiu Florentin Stoenica** [5]

1   Faculty of Economics and Business Administration, Dimitrie Cantemir Christian University, 040042 Bucharest, Romania

2   Faculty of Finance-Banking, Accountancy and Business Administration, Titu Maiorescu University, 040051 Bucharest, Romania; sorinelcapusneanu@ucdc.ro

3   Faculty of Economic Sciences, 1 Decembrie 1918 University, 510009 Alba-Iulia, Romania; dan.topor@uab.ro (D.I.T.); mihaela.hint@uab.ro (M.Ș.H.)

4   National Institute for Chemical Pharmaceutical Research and Development (INCD-ICCF), 031299 Bucharest, Romania; starasadela@gmail.com

5   Faculty of Management—Marketing, Artifex University of Bucharest, 060754 Bucharest, Romania; lstoenica@artifex.org.ro

*   Correspondence: mirela.turkes@ucdc.ro; Tel.: +40-72-817-6475

**Abstract:** The purpose of our study is to research and identify intrinsic and extrinsic motivations but also their impact on the behavioral intentions of using Internet of Things (IoT) solutions among company managers in Romania in the future. The research method used in the quantitative study was the sample survey, using the online questionnaire as a data collection tool. The questionnaire included a formalized set of 54 questions, being specially designed to generate a new structural model starting from the Unified Theory for Acceptance and Use of Technology (UTAUT) elements. A total of 416 respondents provided complete and useful answers for this research. The results of the study showed that the endogenous factor intrinsically motivated significantly and strongly affected the intention of managers to use IoT solutions in professional activity compared to the variable of extrinsic motivation whose action remains insignificant. At the same time, the effect of the intrinsic motivation variable was mediated by the significant influence of the endogenous extrinsic motivation factor. The implications of this study are multiple for both managers and researchers. The originality of this article lies in the empirical part of the research, which, by using a quantitative method based on the questionnaire, provides important information on the impact of the motivational spectrum on the acceptance and use of IoT solutions by managers in the next period to achieve new performance, both personal and professional.

**Keywords:** IoT; UTAUT; intrinsic motivation; extrinsic motivation; perceived privacy; perceived security; behavioral intention; digital era

## 1. Introduction

Technology-based innovation is the main support in the efficient implementation and transformation of present business processes [1]. With the development of companies, the digital transformation of business processes and models has increased in order to gain competitive advantages through the use of the Internet of Things (IoT). Connecting anytime and anywhere with IoT solutions is done through a vast network of electronic devices containing sensors or software [2], connected people, connected networks, connected data and connected processes that have created revolutionary opportunities in the economy, society, business and personal life [3]. It is estimated that by 2030 there will

be over 500 billion devices connected to the Internet [4]. However, the availability of IoT technologies and solutions does not guarantee acceptability from end users. As a result, understanding user acceptance becomes an essential condition for expanding the use of IoT applications in everyday life.

Although users enjoy multiple benefits [5–10], the proliferation of new communication technologies has raised privacy and security issues due to the unprecedented amounts of information it can collect, analyze and store [11]. Researchers have conducted studies on the possible implications of IoT services on users' lives, and as a result, understanding user acceptance becomes an essential condition in the daily expansion of IoT applications. As IoT development is still in its infancy, few studies have been devoted to understanding acceptance issues from end users and especially from the perspective of company managers. Some researchers have theoretically investigated IoT through the prism of users, governments, and companies [12–15], implementation, architecture, and design [16,17]. Other researchers have focused in their studies on investigating the negative aspects of IoT such as identification, heterogeneity, addressing and interoperability [12] or lack of acceptance of user acceptance as an obstacle in IoT adoption and deployment [18].

As the development of IoT is still at an early stage, few studies have been dedicated to understanding the problems of acceptance by end users and especially from the perspective of company managers. Given the importance given to new technologies in the digital age but also the difficulties of understanding and acceptance faced by users, this paper aims to develop and empirically test a model of the causal relationship of intrinsic and extrinsic factors that determine acceptance of IoT by managers companies. The decisional problem is described by the question: To what extent do specific elements of motivation influence the behavioral intention of Romanian managers related to the use of IoT solutions in professional activity? The purpose of marketing research was to identify factors in the spectrum of intrinsic and extrinsic motivations and determine their impact on the behavior of using IoT solutions among company managers in Romania in the future.

The objectives of our research aim at: (1) Assessing the impact of the variables: Effort Expectancy, Attitude Towards IoT, Perceived Autonomy and Perceived Competence and Perceived Relatedness on the spectrum of intrinsic motivation; (2) Determining the direct impact of the factors: Performance Expectancy, Social Influence, Facilitating Conditions, Perceived Privacy and Perceived Risk/Security on extrinsic motivation; (3) Determining the way in which intrinsic and extrinsic motivation influences the behavioral intention to use IoT solutions by the managers of Romanian companies at work.

This study intends to make several contributions such as: (1) researching and identifying the intrinsic and extrinsic motivations of company managers in accepting and using IoT solutions; (2) the extension of the specialized literature by determining the impact of the spectrum of motivations on the behavior of future use. Previous studies have not highlighted the assessment of behavioral intent regarding the future use of IoT solutions by company managers but only the identification of factors that influence the acceptance of the IoT system in public universities [19].

The content of the article is structured in accordance with the proposed purpose and objectives, with a brief presentation of the literature and research methodology and data collection and processing in Section 2, while the empirical results are presented and discussed in Section 3. Finally, the conclusions and limitations of the study are presented in Section 4.

## 2. Materials and Methods

### 2.1. Materials

#### 2.1.1. IoT: Definition and Evolution

IoT was introduced in 1999 by Kevin Ashton and is the network that connects objects in the physical world of the Internet, allowing communication between them, collecting and exchanging data between them and people in order to achieve goals in different fields and applications [20]. Developed based on RFID technology by MIT's Auto-ID Center, IoT has received several definitions: (1) focusing on smart infrastructure with Radio Frequency Identification (RFID) technology [21], (2) ubiquitous

connection [22], (3) dynamic global network infrastructure with the identification of physical and virtual things [23], (4) a network containing all smart devices with a kind of detection mechanism that can communicate via the Internet with other smart or cloud devices [24].

Smart devices are those devices that have built-in systems with Internet connectivity, being equipped with a microcontroller, sensors and actuators. The sensors receive the real-world data, the microcontroller processes this data and the actuators create the events based on them. These devices are self-configured and can be controlled remotely [25–27]. IoT components consist of: (1) sensors, RFID technology and near field communication (NFC) with short-term wireless data collection technologies, (2) wireless or wired Internet networks for data transmission, and (3) applications of services for processing, analyzing and managing transmitted data [28]. In other words, IoT is a vast computing network that connects wireless sensors to application services over the Internet.

Whether we are talking about employees or managers, studies on the behavior of users of IoT solutions are limited in the academic literature and are still in an early stage [29]. Adopting new technology can bring benefits (convenience, speed in decision making) but also negative effects (anxiety or technological stress) to its users [30]. Thus, three types of paradoxes with double impact on users of computer products were identified: (1) control/chaos; (2) meeting needs/creating needs; (3) freedom enslavement [31–33].

The increase in the number of users (corporations, companies, SMEs, individuals) who use IoT can be explained by the many benefits it offers such as: (1) improving the customer experience; (2) improving efficiency; (3) access to data/collection of new data; (4) reduction of labor cost/actual costs; (5) connectivity; (6) saving time; (7) a new positioning at the business level [34]. Although the benefits to users of IoT applications are obvious, there are a number of concerns about: (1) complexity, (2) dependence on other IoT applications, (3) business models used, (4) society, (5) regulatory standards and (6) data security/protection (unauthorized access to personal information and risks of hacking on IoT applications) leading to positive factors that promote IoT use and negative factors that prevent IoT use [34,35].

### 2.1.2. Unified Theory for Acceptance and Use of Technology (UTAUT): Importance, Advantages and Limitations

The clearest highlighting of the relationships between the degree of acceptance of information technology and IoT by employees, employers or managers and their intention to use information technology is achieved using the Unified Theory for Acceptance and Use of Technology (UTAUT). Based on the general model of technology adoption, the UTAUT model was developed as an integration of previous research models in the field of technology acceptance [36]. This model has incorporated several theories such as: rational action theory (TRA) [37], acceptance model technology (TAM) [38], TAM 2 [39], motivational model (MM) [40], planned behavioral theory [41], C-TAM-CTB model, computer use model (MPCU) [42], social cognitive theory (SCT) [43] and the theory of innovative diffusion (IDT) [44].

In its construction, the UTAUT model uses four basic variables to reflect the above-mentioned relationships: Performance Expectancy (PE) (users' perception of the usefulness of IoT in their professional activity), Effort Expectation (EE) (ease perceived by users in using IoT), Social Influence (SI) (perception of social pressure in IoT use) and Facilitating Conditions (FC) (user perception of support for IoT use) to make predictions about user Behavioral Intentions (BI) and Use Behavior (UB) in adopting IoT for professional purposes in the workplace. The UTAUT model also uses the following variables: gender, age, previous experiences with IoT or other technologies, the degree of volunteering and influencing independent variables (PE, EE, SI, FC) and dependent variables (BI, UB).

Like any other model, UTAUT has advantages and limitations. According to specialists, the most important advantages of the UTAUT model are: (1) the holistic approach in explaining the basic relationships with social and psychological factors with a significant impact on information technology; (2) consistent validity and reliability of collected data [45,46]. Among the limits of the UTAUT model identified by the most significant specialists are: (1) the multitude of variables necessary to achieve the substantial level of variance; (2) due to the complexity of SI and FC they cannot be measured correctly [47].

Considering the advantages and limitations of the UTAUT model, the specialists tried to investigate the acceptance of information technology by analyzing the intrinsic and extrinsic motivations of the staff employed to obtain feedback for the design of effective strategies by companies for promotion in the workplace [48].

### 2.1.3. Motivation and Relationships between Intrinsic and Extrinsic Motivations of Employees and Managers

Work motivation is defined as "the psychological forces that determine the direction of a person's behavior in an organization, a person's level of effort, and a person's level of persistence in the face of obstacles" [49]. In other words, work motivation can be approached through the prism of motivation: (1) intrinsic (refers to noninstrumental behavioral involvement that is inherently satisfactory and pleasant, motivated action not depending on the outcome of the behavior) and (2) extrinsic (refers to performance instrumental behavior that is fundamentally contingent on achieving a result separate from the action itself) [50]. In the professional environment, employees as well as managers would be intrinsically motivated by pleasure and satisfaction in performing their tasks [51,52] but also personal performance [53]. Extrinsic motivation is related to objectives such as financial rewards or incentives obtained [51,52]. Defined from the practical point of view, motivation guides the direction, intensity and persistence of performance behaviors [53]. Generally speaking, extrinsic and intrinsic motivations influence the intentions of employees or managers regarding activities and behaviors but the factors that influence these motivations are very little known [51]. The studies conducted by specialists explored the motivations of employees from both private and public, and how managers perceived these influences of intrinsic and extrinsic factors on employee involvement were reflected in the following results: (1) monetary rewards are more important in motivating private sector employees while job security has been more important in motivating public sector employees, both categories of employees equally appreciated the desire to work in a team, contribution to society and opportunities for advancement [54]; (2) private sector employees appreciate higher extrinsic rewards but also intrinsic variables compared to public sector employees who appreciate higher intrinsic rewards in light of motivation [55]; (3) public and private employees have similar attitudes about job security, promotion opportunities, and the significance of work, but private employees value cash rewards more than public employees who value shorter working hours more [56].

### 2.1.4. Intrinsic and Extrinsic Motivational Factors

According to our study, we considered intrinsic motivational factors: Effort Expectancy (EE), Attitude Towards IoT (IoTA), Perceived Autonomy (PA), Perceived Competence (PC), Perceived Relationship (PR). The extrinsic motivational factors considered were: Performance Expectancy (PE), Facilitation Conditions (CF), Social Influence (SI), Perceived Confidentiality (PP), Perceived Security (PS) and Behavioral Intention (BI).

Effort Expectancy (EE) shows the extent to which the use of technology is easily perceived by users [36] and which also includes the degree of perceived complexity [57] being important in the context of personal and workplace technologies. Perceptions of ease of use [38] and perceived utility have been reflected in several studies using the TAM model with a significant impact on behavioral intentions to use IoT technologies such as: (1) e-learning [58], (2) blogs [59], (3) mobile devices [60], (4) IoT applications [61], (5) wearable technology [62,63], (6) adoption of mobile banking [64] and Internet

banking [65], (7) urban technologies [66], (8) virtual reality [67] and (9) teaching technologies [68]. Therefore, we consider that the performance effort will have a positive impact on the intrinsic motivation of using IoT solutions by managers and the first hypothesis that will be tested will be:

**H1.** *Effort Expectancy has a significant impact on Intrinsic Motivation*

Attitude Towards IoT (IoTA) shows the extent to which users express their feelings or opinions about IoT solutions. Specialist studies have shown positive attitudes of most users related to IoT solutions [69] finding them interesting, captivating and attributing a higher quality to the information transmitted [70] thus contributing to efficiency, safety and convenience to their activities. Based on the robustness of IoT applications that contribute to facilitating the activities of managers, the interest, pleasure and joy of using them by managers, we believe that the attitude towards IoT will have a positive impact on the intrinsic motivation to use IoT solutions by managers. and the second hypothesis tested will be:

**H2.** *Atitudine toward IoT has a significant impact on Intrinsic Motivation*

Perceived Autonomy (PA) shows the degree to which the user of a particular IoT system considers that he would use it with the minimum necessary resources (e.g., additives, supplements, accessories, additional hardware, additional software, extra networking and additional energy) [71] in order to carry out managerial activities. Considering the different options and opportunities for developing the desired activities which will allow managers to better coordinate their activities by finding efficient and fast solutions, while monitoring and controlling a wide range of IoT devices, the third hypothesis was formulated. It consists of the manifestation of a positive impact of the perceived autonomy on the intrinsic motivation of using IoT solutions by managers.

**H3.** *Perceived Autonomy has a significant impact on Intrinsic Motivation*

Perceived Competence (PC) shows the ability of a person to successfully perform the tasks received being significantly correlated with the results obtained [72]. Correlated with perceived autonomy and self-efficacy, self-competence becomes an important predictor for the learning environment [72,73] being strongly related to intrinsic motivation [74]. Users' perception of IoT solution management skills, skills developed through connectivity to IoT solutions and their use, we consider that the perceived competence has a positive impact on the intrinsic motivation to use IoT solutions by managers and the fourth hypothesis tested will be:

**H4.** *Perceived Competence has a significant impact on Intrinsic Motivation*

Perceived Relatedness (PR) indicates the quality of the relationship between the user of IoT solutions and the perception given by the user. Based on the theory of self-determination, perceived relationship plays an important role along with competence and autonomy [75] but also with perceived competence in motivating users [76]. Therefore, it is expected that the support, understanding, listening and appreciation received from other users of IoT solutions will create a positive impact on the intrinsic motivation to use IoT solutions by managers and the fifth hypothesis tested will be:

**H5.** *Perceived Relatedness has a significant impact on Intrinsic Motivation*

Performance Expectancy (PE) shows the extent to which the use of technology brings benefits to users [36] being one of the important predictors of the behavioral intent of communication technologies [77]. The more individuals expect to improve their performance through the use of technology, the more likely they are to use it. The use of new technologies brings many advantages being useful in daily activities, allowing the acquisition of interpersonal skills and helping to quickly perform work tasks, thus increasing labor productivity. The same model of performance expectation is expected to have a positive impact on the extrinsic motivation of managers to use IoT solutions, thus testing the sixth hypothesis:

**H6.** *Performance Expectancy has a significant impact on Extrinsic Motivation*

Social Influence (SI) shows the extent to which the use of technology is perceived through the prism of those close to them [36] having special importance [78] especially through the significant influence on the intention to the adoption of IoT solutions [79]. Social influence is "the perception of users if people important to them perceive that they should be involved in behavior" [80] and is especially important for consumers, who do not have much detailed information about the use of products and services recently launched and which can reach reliable information through social interaction [61]. Through the support provided, managers receive help, advice and useful information related to IoT solutions, becoming more confident in making independent decisions and guaranteeing partnerships in providing IoT solutions. Starting from the above, we consider that social influence will have a positive impact on the extrinsic motivation of managers to use IoT solutions, thus testing the seventh hypothesis:

**H7.** *Social Influence has a significant impact on Extrinsic Motivation*

Facilitating Conditions (FC) show the extent to which the use of technology is perceived by users in terms of available resources and support provided [36] to perform a behavior. In addition to facilitation conditions, specialists used in their studies other variables such as perceived value, self-efficacy [81] and the results indicated and supported a relationship between facilitation conditions and social assistance in the use of technology. Benefiting from IT consulting, users have the knowledge and skills to use the IoT solutions needed to exchange ideas in the IoT partner community. We consider that the facilitation conditions also have a positive impact on the extrinsic motivation of managers to use IoT solutions, thus testing the eighth hypothesis:

**H8.** *Facilitating Conditions has a significant impact on Extrinsic Motivation*

Perceived Privacy (PP) is defined as the extent to which a user using private IoT solutions considers that they could exercise some control over how information used by the system can be accessed, collected, stored, used, manipulated or disclosed to other systems or users [71]. Confidentiality is an important challenge in the IoT environment, due to the availability of sensory devices, the speed and volume of information flow [72]. By using strong passwords to maintain control over IoT solutions, certain vulnerabilities in data networks can be eliminated. We consider that the perception of confidentiality has a positive impact on the extrinsic motivation of managers to use IoT solutions, thus testing the ninth hypothesis:

**H9.** *Perceived Privacy has a significant impact on Extrinsic Motivation*

Perceived Security (PS) indicates the degree to which a user of an IoT system considers that it would be protected from unauthorized access to the modification, destruction or disclosure of its resources or personal information during the processing, storage and transmission of data [71]. Since most IoT devices are wireless, this involves many security challenges such as forgery, network attacks, denial of service, intrusions, and so on [82–85] which causes a lot of damage [86]. Solutions have also been identified to increase trust in IoT devices by increasing security in business management by using symmetric keys, encryption or other ways to secure information [87]. We consider that the perception of security has a positive impact on the extrinsic motivation of managers to use IoT solutions, thus testing the tenth hypothesis:

**H10.** *Perceived Security has a significant impact on Extrinsic Motivation*

Behavioral Intention (BI). Specialists obtained positive results regarding the analysis of consumers' intentions to buy IoT products using several factors such as: (1) connectivity, interactivity, telepresence, intelligence, convenience and security using functional experience as a variable [88]; (2) behavioral control, perceived pleasure [61]. We consider that extrinsic motivation has a significant impact on the behavioral intention of managers to use IoT solutions, thus testing the eleventh hypothesis:

**H11.** *Extrinsic Motivation has a significant impact on Behavioral Intention to use IoT solutions*

Intrinsic Motivation (IM) is defined as the performance of an activity for its inherent satisfaction (pleasure), rather than for a separate outcome [53], which reflects man's natural willingness to assimilate and learn [75]. Specialist research has indicated the following increased intrinsic motivations related to: (1) employees' willingness to create a positive mood, which leads to increased learning and knowledge exchange [51]; (2) strong motivation for self-learning and self-realization from a professional point of view [89]; (3) perception as a safer result of performing a task than in the case of extrinsic reasons being the only functional engine of performance [53,90]. We consider that intrinsic motivation has a significant impact on the behavioral intention of IoT solution users, thus testing the twelfth hypothesis:

**H12.** *Intrinsic Motivation has a significant impact on Behavioral Intention to use IoT solutions*

Extrinsic Motivation (EM) is defined as obtaining different results when performing an activity [75]. Achieving goals through extrinsic motivation consists of obtaining organizational rewards or benefits [51] but this behavior can only be temporary. The extrinsic rewards between the two motivators will be more relevant and can replace intrinsic motivation as the main goal for the involvement of a particular activity [91]. We consider that extrinsic motivation has a significant impact on the intrinsic motivation of managers to use IoT solutions, thus testing the thirteenth hypothesis:

**H13.** *Extrinsic Motivation has a significant impact on Intrinsic Motivation*

*2.2. Methods*

Quantitative marketing research aims to identify the opinions and motivations of Romanian managers on the use of IoT solutions, given the development of new skills needed in future professional activities. The initial UTAUT model included four constructions: performance expectation, effort expectation, social influence and facilitation conditions, being ideal in predicting the intention to accept and use new technologies and identify differences between users' intentions to use an information system and subsequent use behavior [36]. Recent studies have developed and strengthened the UTAUT model and highlighted positive relationships between new elements and the intention to use the new IoT concept [19,92,93]. Most previous research on the influence of intrinsic and extrinsic motivation on behavioral intention to use has focused more on issues of autonomy, control, performance, and facilitation [10,40,50,53]. However, no study has analyzed the relationship between intrinsic and extrinsic motivation variables and the intention to use IoT solutions in the future by different types of managers. However, no study has analyzed the relationship between intrinsic and extrinsic motivation variables and the intention to use IoT solutions in the future by different types of managers. Intrinsic and extrinsic motivations that determine managers to engage in competition, to develop new working relationships, to get actively involved in work and achieve new performances, correlated with maintaining the confidentiality and security of personal data.

The research model proposed in Figure 1 divides the factors of the UTAUT model into two groups, joining new elements specific to motivation. Thus, in the first stage of the model is presented the action exerted by the exogenous variables Effort Expectancy (EE), Attitude Towards IoT (AT), Perceived Autonomy (PA), Perceived Competence (PC) and Perceived Relatedness (PR) on the Intrinsic Motivation factor (IM). In the second stage, Extrinsic Motivation (EM) knows the influence of latent factors: Performance Expectancy (PE), Social Influence (SI), Facilitating Conditions (FC), Perceived Privacy (PP) and Perceived Security (PS). The last stage shows the impact of exogenous variables intrinsic motivation and extrinsic motivation on Behavioral Intention to use IoT solutions (BI).

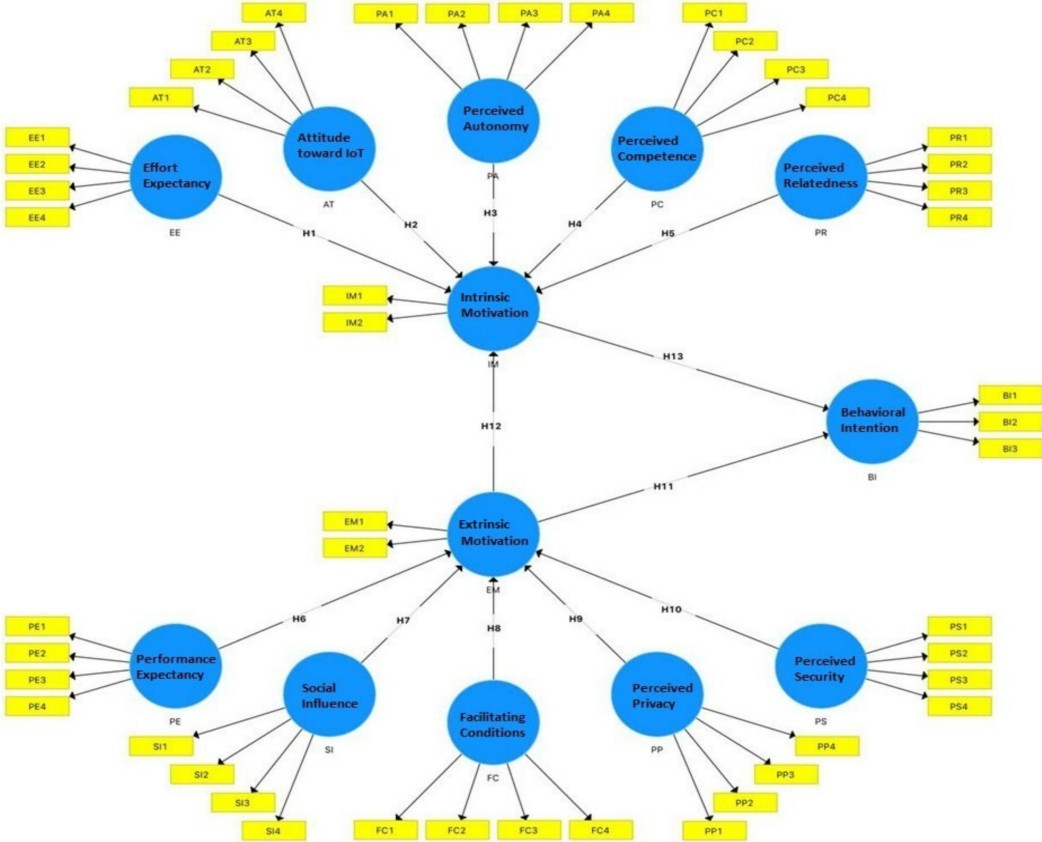

**Figure 1.** Model construct.

### 2.2.1. The Questionnaire

The questionnaire included a formalized set of 54 questions (47 content questions and 7 identification questions), specially designed to generate a new multidimensional model starting from the UTAUT elements [36]. Figure 1 shows the articles for the 13 latent variables included in the construct model. Effort Expectancy (EE) articles were adopted. Attitude Towards IoT (AT), Perceived Autonomy (PA), from the studies of authors Chen (2013), Economides (2017) and Wong (2002), while various new articles were introduced for Perceived Competence (PC) and Perceived Relatedness (PR) from Ryan (2000) and Karahanna (1999), respectively. For Performance Expectancy (PE), Social Influence (SI) and Facilitating Conditions (FC) were adapted from the research of Davis (2003), Venkatesh (2012) and Economides (2017). Other new measurement elements were introduced for Perceived Privacy (PP) and Perceived Security (PS) from material from Albalawi (2018) and Dong (2014). The items that measure Intrinsic Motivation (IM) and Extrinsic Motivation have been adapted for managers who intend to use IoT solutions in the future. Gender, age, education level, management level, work experience, industry and location were the moderators used in our study. Respondents were asked about their work experience, managerial position and plans or intentions to use IoT solutions in the future. The use of a self-administered online questionnaire required the construction of short, closed questions that were subsequently measured using the five-level Likert scale, from "1"—Strongly disagree to "5"—Strongly agree to "1"—Strongly disagree to "5"—Strongly agree.

### 2.2.2. Sample Selection

The testing of the thirteen working hypotheses involved the use of a known marketing method, namely the sample survey, which took place between 10 February and 30 June 2020. With a questionnaire response rate of approximately 34%, the sample finally included 416 respondents (62.74% men and 37.26% women), 48.80% of the subjects coming from the middle-level managers (general managers,

branch managers and department managers) while only 14.42% belonging to the group of top-level managers (the board of directors, president, vice-president or CEO). Almost half of the respondents (48.32%) had more than 11 years of work experience.

They operate in various fields: discrete and process manufacturing (32.45), retail (24.28%), transportation and logistics (9.13%), automotive and consumer electronics (5.53%) and others (Table 1).

**Table 1.** Sample structure.

|  | **Dimensions** | **Frequency** | **Percentage** |
|---|---|---|---|
| Gender | Male | 261 | 62.74 |
|  | Female | 155 | 37.26 |
| Age | Between 20 and 30 years | 53 | 12.74 |
|  | Between 31 and 40 years | 104 | 25.00 |
|  | Between 41 and 50 years | 151 | 36.30 |
|  | Between 51 and 60 years | 80 | 19.23 |
|  | Upper 60 years | 28 | 6.73 |
| Education Level | Primary education | 18 | 4.33 |
|  | High school | 60 | 14.42 |
|  | University | 214 | 51.44 |
|  | Master/PhD | 119 | 28.61 |
|  | Other | 5 | 1.20 |
| Management Level | Top-Level Management | 60 | 14.42 |
|  | Middle-Level Management | 203 | 48.80 |
|  | Lower-Level Management | 153 | 36.78 |
| Work Experience | Less than 5 years | 28 | 6.73 |
|  | Between 5 and 10 years | 187 | 44.95 |
|  | Between 11 and 16 years | 129 | 31.01 |
|  | More than 16 years | 72 | 17.31 |
| Industry | Automotive and Consumer Electronics | 23 | 5.53 |
|  | Healthcare and Hospitality | 12 | 2.88 |
|  | Retail | 101 | 24.28 |
|  | Discrete and Process Manufacturing | 135 | 32.45 |
|  | Finance services | 16 | 3.85 |
|  | Energy and Natural Resources | 7 | 1.68 |
|  | Communications | 13 | 3.13 |
|  | Transportation and Logistics | 59 | 14.18 |
|  | Construction and Utilities | 38 | 9.13 |
|  | Agriculture/Agrotech/farming | 9 | 2.16 |
|  | Other | 3 | 0.72 |
| Location | AB | 7 | 1.68 |

**Table 1.** *Cont.*

| Dimensions | Frequency | Percentage |
|:---:|:---:|:---:|
| AR | 10 | 2.40 |
| AG | 12 | 2.88 |
| BC | 9 | 2.16 |
| BH | 6 | 1.44 |
| BN | 7 | 1.68 |
| BT | 4 | 0.96 |
| BV | 3 | 0.72 |
| BR | 11 | 2.64 |
| B | 114 | 27.40 |
| BZ | 5 | 1.20 |
| CS | 6 | 1.44 |
| CL | 2 | 0.48 |
| CJ | 17 | 4.09 |
| CT | 11 | 2.64 |
| CV | 6 | 1.44 |
| DB | 5 | 1.20 |
| DJ | 9 | 2.16 |
| GL | 13 | 3.13 |
| GR | 5 | 1.20 |
| GJ | 1 | 0.24 |
| HR | 3 | 0.72 |
| HD | 4 | 0.96 |
| IL | 9 | 2.16 |
| IS | 5 | 1.20 |
| IF | 27 | 6.49 |
| MM | 9 | 2.16 |
| MH | 3 | 0.72 |
| MS | 8 | 1.92 |
| NT | 5 | 1.20 |
| OT | 7 | 1.68 |
| PH | 16 | 3.85 |
| SJ | 1 | 0.24 |
| SM | 3 | 0.72 |
| SB | 13 | 3.13 |
| SV | 5 | 1.20 |
| TR | 2 | 0.48 |
| TM | 15 | 3.61 |
| TL | 3 | 0.72 |
| VS | 7 | 1.68 |
| VL | 2 | 0.48 |
| VN | 6 | 1.44 |
| Total | 416 | 100 |

### 2.2.3. Methodology and Data Analysis

The collection of data and information necessary for the research was carried out using a questionnaire distributed by email to managers working in Romania. It was decided to use the online survey due to the multiple advantages offered: obtaining relevant information for marketing research in a convenient time, studying a large sample, low costs, convenience for the respondent (manager), flexibility in link identification that did not reach the email addresses from the initial list and the ease of collecting and processing the data necessary for the research [94]. The researched population was identified in the databases of the Romanian Chamber of Commerce and Industry, Romanian Entrepreneur Association and The Romanian Academic Society of Management (RASM), etc. In order to carry out the quantitative research in optimal conditions, a sampling base was built which included 1224 managers, each observation unit being randomly selected from the reference

population. After being contacted and expressing their willingness to participate in the marketing research, managers completed the online questionnaire (Table 2).

**Table 2.** Questionnaire.

| Items | Questions |
|---|---|
| Effort Expectancy (EE) | (EE1) Learning how to use IoT solutions would be easy for me<br>(EE2) I think I could easily use IoT solutions<br>(EE3) My interaction with IoT solutions would be clear and easy to understand<br>(EE4) I think it would be easy for me to become proficient in monitoring, controlling and using IoT solutions |
| Attitude Towards IoT (AT) | (AT1) Using robust IoT solutions will make my job a lot easier<br>(AT2) I am interested in understanding and using IoT solutions<br>(AT3) I like to use IoT solutions because they involve many challenges<br>(AT4) I am happy when I use IoT solutions because it allows me to monitor and control the ecosystem |
| Perceived Autonomy (PA) | (PA1) Using IoT solutions, I will have new options and opportunities for developing the desired activities<br>(PA2) I feel that using IoT solutions will allow me to better coordinate activities at work<br>(PA3) Using IoT solutions can reduce the number of decisions I have to make to get optimal results<br>(PA4) By using IoT solutions, I have the freedom to monitor and control a wide range of devices remotely |
| Perceived Competence (PC) | I think I'm pretty good at connecting devices and using IoT solutions<br>I think I have the skills to build IoT solutions if I attend specific courses;<br>I think I am quite good at getting involved in a programming project on the development of IoT solutions in the future<br>I think I'm good at managing IoT solutions |
| Perceived Relatedness (PR) | (PR1) When I use IoT solutions, I feel closer to my colleagues and business partners<br>PR2 When I use IoT solutions, I feel better understood by colleagues and business partners<br>PR4 When I use IoT solutions, I am better listened to by colleagues and business partners<br>PR4 When I use IoT solutions, I feel more connected to colleagues and business partners |
| Intrinsic Motivation (IM) | (IM1) When I use IoT solutions, I feel involvement, pleasure and satisfaction<br>(IM2) When I use IoT solutions, I can easily master any task |
| Performance Expectancy (PE) | (PE1) IoT solutions are useful both for my daily life and for my job<br>(PE2) IoT solutions help me quickly accomplish work tasks<br>(PE3) IoT solutions would increase the productivity of my work<br>(PE4) IoT solutions allow me to acquire new interpersonal skills in programming with a direct impact on increasing my morale |
| Facilitating Conditions (FC) | (FC1) To use IoT solutions, they have all the necessary resources: skills and knowledge<br>(FC2) I have the necessary knowledge to use IoT solutions (IoT Developer Guide and/or exploring classical/online learning modules)<br>(FC3) Using IoT solutions are useful for exchanging ideas in the IoT partner community<br>(FC4) I benefit of IoT consulting and technical support |
| Social Influence (SI) | (SI1) People who are important to me have given me useful tips and information about using IoT solutions<br>(SI2) People who are important to me have helped me find the best IoT solutions<br>(SI3) People who are important to me have given me the support I need to use IoT solutions<br>(SI4) People who are important to me believe that I have the skills to become a partner in providing IoT solutions |
| Perceived Privacy (PP) | (PP1) When I use IoT solutions, I feel that other systems would have control over my work<br>(PP2) When I use IoT solutions, I feel that there are certain risks when it comes to manipulating information<br>(PP3) When I use IoT solutions, from reputable manufacturers I feel that my information is safe/secure<br>(PP4) When I use strong password-protected IoT solutions, I have the security of my information |
| Perceived Security (PS) | (PS1) When I use IoT solutions, I feel protected by unauthorized access to other systems<br>(PS2) When I use IoT solutions, additional methods of verifying my access are used<br>(PS3) When I use IoT solutions, I feel protected by the use of encryption technologies<br>(PS4) When I use IoT solutions, the website has a data security policy through which it assumes responsibility |
| Extrinsic Motivation (EM) | (EM1) When I use IoT solutions, I feel that I contribute to my success<br>(EM2) When I use IoT solutions, I feel the recognition of personal achievements |
| Behavioral Intention (BI) | (B1) I intend to use IoT solutions in the future<br>(B2) I plan to use IoT solutions in the future/year<br>B3) I foresee the use of IoT solutions in the future |

The processing, analysis and interpretation of data and information provided by managers was performed using the software SPSS Statistics and SmartPLS3 Professional. To examine the structural model and measure the simultaneous influence between its latent variables, the collected data were tabulated in SPSS and analyzed using the structural equation modeling technique based on the partial least squares (PLS). Two major advantages were the basis for the option to use this exploratory technique, namely: fast and correct validation of reflective construct models [95] and increased efficiency compared to other regression techniques [96].

### 2.2.4. Questionnaire Validation and Preliminary Results

Because the success of any marketing research depends overwhelmingly on the quality of the questionnaire in its design, two criteria were taken into account: ensuring the management of the information necessary to substantiate the decisions and the characteristics of the respondents. The model obtained from the design process was analyzed by a team of specialists who have solid knowledge, are involved in IoT projects and support services. After eliminating the double, confusing or unnecessary questions, its pretesting was performed in all aspects, on a restricted sample of respondents, the best data operators and respecting the proposed methodology. The results of the pretesting of the questionnaire were analyzed very carefully, bringing the necessary improvements. After collection, the data were entered into the SPSS application to verify the correctness of the information provided by the respondents but also to eliminate the errors. Interactive data validation involves validating the questionnaire as soon as corrections have been made. Preliminary results (215 respondents in the previous period) for the latent variables—Attitude Towards IoT (CA = 0.839, AVE = 0.682), Perceived Autonomy (0.752, AVE = 0.673, Perceived Privacy (CA = 0.746, AVE = 0.639), Social Influence (CA = 0.773, AVE = 0.697), Intrinsic Motivation (CA = 0.783, AVE = 0.814)— indicated measures Cronbach's alpha (CA) greater than 0.6 and Average Variance Extracted (AVE) greater than 0.5 [97]

### 3. Results and Discussions

For the proposed internal model, the values of the path coefficients were positive (Figure 2) except for those from Perceived Competence (PC) to Intrinsic Motivation (IM) and from Perceived Privacy (PP) to Extrinsic Motivation (EM) whose weights reflected negative paths of −0.010 and −0.004, respectively.

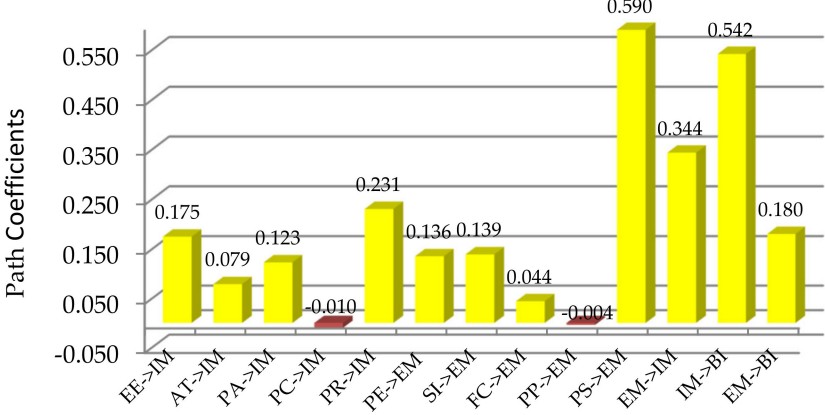

**Figure 2.** Path coefficients for the inner model.

Being always standardized, the weights of the coefficients register variations between −1 and +1, their weights reflecting weaker or stronger ways [98]. Three strong effects are visible among the paths from Perceived Risk (PS) to Extrinsic Motivation (EM) (0.590), from Extrinsic Motivation (EM) to Intrinsic Motivation (IM) (0.344) and from Intrinsic Motivation (IM), respectively, to the Behavioral Intention (BI) (0.542).

In Figure 3 along with the standardized path coefficients, placed on the corresponding paths, the values of the $R^2$ value for the three endogenous latent variables (factors) were displayed in the blue ellipses. The general measure of the effect size for the structural model, the $R^2$ can register values "high" (R > 0.5), "moderate" (R > 0.30) or "weak" (R > 0.1) [98]. For the endogenous variable Behavioral Intention to Use Solutions IoT the value of the $R^2$ indicates a high size (0.452), which means that almost 45.2% of its variation is explained by the model, by the common action of two factors (IM and EM). Five exogenous variables (EE, AT, PA, PC and PR) jointly explain approximately 53.8% of the variation of the endogenous Intrinsic Motivation (IM) factor. In the case of the Extrinsic Motivation (EM) factor, the value of the $R^2$ was more substantial, with 63% of the variation of the endogenous variable being explained by five other exogenous variables (PE, SI, FC, PP and PS).

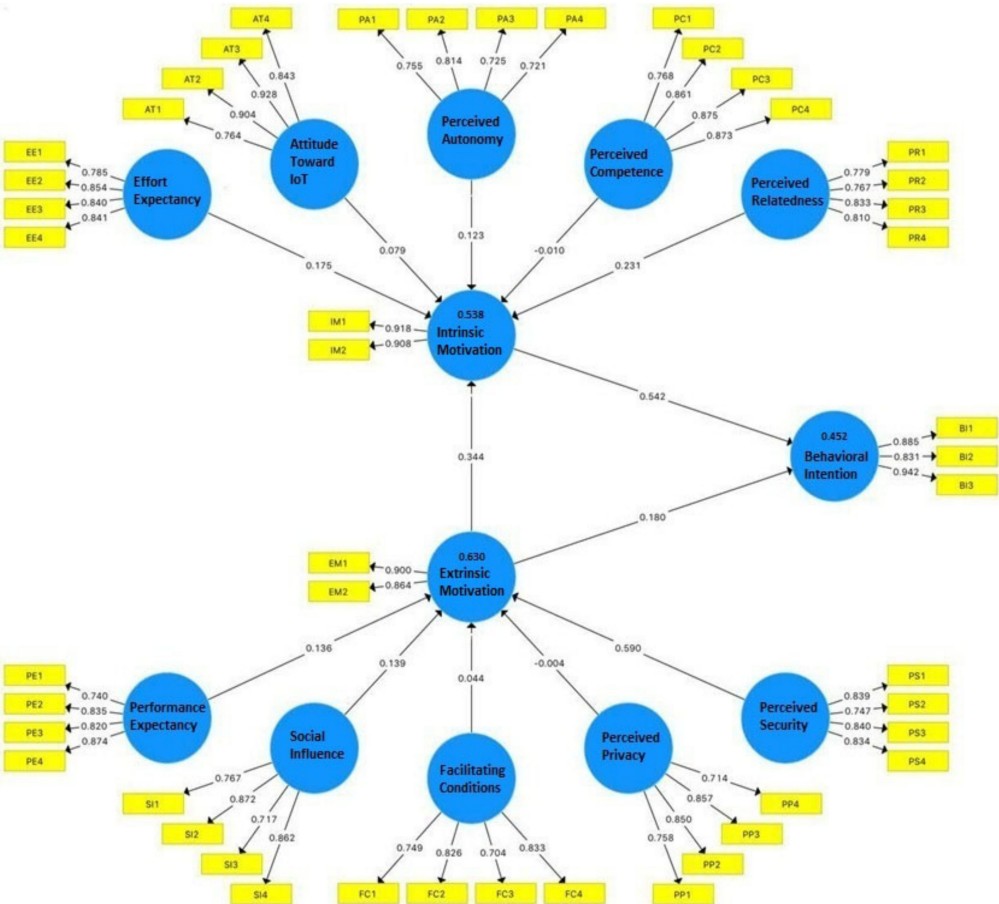

**Figure 3.** Partial least squares (PLS) algorithm results.

Measurement loads linking factors to indicator variables should generally be significant and range from 0 to 1 [99]. The proposed reflective model is strong and reliable, confirmed by the size of the standardized track loads which are greater than 0.70 and by their reliability which is over 0.50. [100]. Calculating the square of the incarnation of the measurements, it is observed that the reliability of all indicators connected to the latent variables EE, AT, PA, PC and PS registers values between 0.52 (0.721 ^ 2) and 0.86 (0.928 ^ 2). At the same time, the absolute contribution of the indicators used to define the latent variables PE, SI, FC, PP and PP indicates high loads, ranging from 0.50 (0.704 ^ 2) to 0.76 (0.874 ^ 2) [101].

### 3.1. The Measurement Model

Table 3 shows the results obtained after testing the convergent and discriminant validity for the construct model. Given that the research focuses on the PLS model, it was decided to test the

convergent validity of the proposed reflective model by using primarily composite reliability, as an alternative to Cronbach's alpha (CA) [100,102].

**Table 3.** Construct reliability and validity.

| Latent Variables | Cronbach's Alpha (CA) | Composite Reliability | Average Variance Extracted (AVE) |
|---|---|---|---|
| AT | 0.800 | 0.872 | 0.639 |
| BI | 0.741 | 0.845 | 0.662 |
| EE | 0.855 | 0.899 | 0.690 |
| EM | 0.717 | 0.875 | 0.778 |
| FC | 0.787 | 0.861 | 0.609 |
| IM | 0.802 | 0.910 | 0.834 |
| PA | 0.757 | 0.841 | 0.570 |
| PC | 0.806 | 0.874 | 0.637 |
| PE | 0.848 | 0.890 | 0.670 |
| PP | 0.771 | 0.856 | 0.602 |
| PR | 0.814 | 0.875 | 0.636 |
| PS | 0.839 | 0.888 | 0.666 |
| SI | 0.727 | 0.823 | 0.542 |

The arguments that strengthened this decision were mainly related to the Cronbach's alpha (CA) indicator which cannot provide a real, perfect estimate of the reliability of the scale-built model. However, the Cronbach's alpha values of each factor were higher than the threshold recommended by Nunnally [103], namely 0.7, so we can say that in this case, it could also confirm the reliability of the measurement model. Previous studies show that the composite reliability of reflective models must be equal to or greater than 0.6 for exploratory studies [103,104] equal to or greater than 0.70 for confirmatory constructs [105] and ≥ 0.80 for confirmatory research [106]. The composite reliability of the proposed construct is very high (> 0.80), a fact confirmed by the size of the indicators obtained by each latent variable. If the composite reliability of the Social Influence factor (0.823) slightly exceeds 0.8, that of the Intrinsic Motivation variable (0.910) approaches 1 (Figure 4).

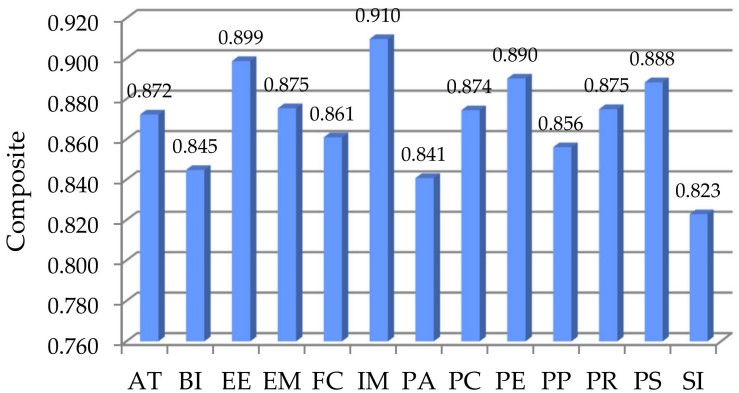

**Figure 4.** Composite reliability.

Although it tends to underestimate reliability, Cronbach's alpha can confirm the convergent validity of a construct model if the indicators for latent variables are ≥ 0.6 [107].

Six latent factors: BI (0.741), EM (0.717), FC (0.787), PA (0.757), PP (0.771) and SI (0.727) were measured at an acceptable level while the rest exceed the ≥ 0.8 limit set for an extremely good scale (Figure 5) [108]. Therefore, the proposed model fulfills the condition of convergent validity.

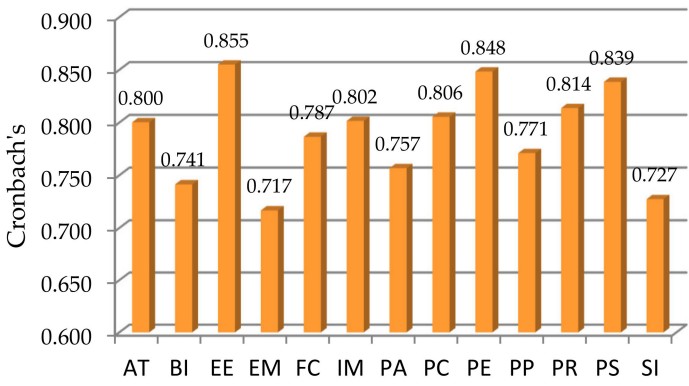

**Figure 5.** Cronbach's alpha.

However, the Average Variance Extracted (AVE) remains the optimal indicator used to test convergent and divergent validity. In the case of reflective models, if the AVE levels of the latency factors exceed the recommended minimum threshold of 0.5 then the convergent validity is confirmed [98,107,109], otherwise, the explained variation is exceeded by the error variation [103,110]. In the present research, the exogenous factors PA (0.570) and SI (0.542) displayed the lowest AVE values but higher than the minimum limit of 0.5. At the opposite pole are the endogenous factors IM and EM whose size AVE was 0.834 and 0.778, respectively. The third endogenous BI factor indicated a moderate AVE of 0.662 (Figure 6). As all AVE values are higher than the minimum acceptable threshold of 0.5, it follows that the construct model meets the condition of convergent validity.

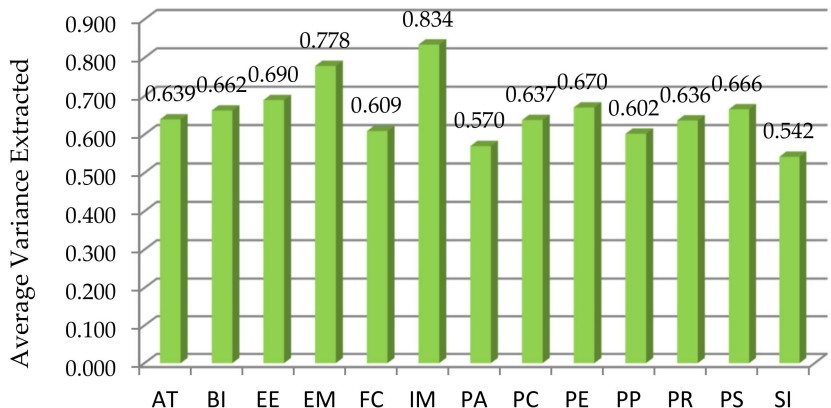

**Figure 6.** Average Variance Extracted (AVE).

The determination of discriminant validity according to the Fornell–Larcker criterion is based on the same AVE indicator [110]. For each factor included in the model, the AVE square roots must be greater than their correlations with any other latent variable [111]. As can be seen in Table 4a,b, the AVE square roots calculated for the 13 factors are distributed in the diagonal cells and the correlations with the other variables are displayed below them.

**Table 4.** Fornell–Larcker criterion and Cross loading.

|     | AT | BI | EE | EM | FC | IM | PA | PC | PE | PP | PR | PS | SI |
|-----|-----|-----|-----|-----|-----|-----|-----|-----|-----|-----|-----|-----|-----|
| AT | 0.800 | | | | | | | | | | | | |
| BI | 0.354 | 0.750 | | | | | | | | | | | |
| EE | 0.408 | 0.442 | 0.830 | | | | | | | | | | |
| EM | 0.394 | 0.529 | 0.503 | 0.882 | | | | | | | | | |
| FC | 0.238 | 0.362 | 0.271 | 0.383 | 0.780 | | | | | | | | |
| IM | 0.421 | 0.558 | 0.555 | 0.543 | 0.282 | 0.913 | | | | | | | |
| PA | 0.261 | 0.365 | 0.244 | 0.320 | 0.351 | 0.372 | 0.755 | | | | | | |
| PC | 0.417 | 0.534 | 0.552 | 0.512 | 0.349 | 0.542 | 0.387 | 0.798 | | | | | |
| PE | 0.472 | 0.506 | 0.583 | 0.563 | 0.342 | 0.574 | 0.258 | 0.539 | 0.819 | | | | |
| PP | 0.239 | 0.419 | 0.374 | 0.491 | 0.262 | 0.358 | 0.252 | 0.596 | 0.515 | 0.776 | | | |
| PR | 0.463 | 0.559 | 0.557 | 0.530 | 0.346 | 0.533 | 0.342 | 0.588 | 0.525 | 0.593 | 0.798 | | |
| PS | 0.372 | 0.564 | 0.594 | 0.577 | 0.427 | 0.475 | 0.289 | 0.515 | 0.582 | 0.508 | 0.563 | 0.816 | |
| SI | 0.236 | 0.382 | 0.406 | 0.473 | 0.296 | 0.330 | 0.257 | 0.354 | 0.380 | 0.288 | 0.358 | 0.458 | 0.736 |

(**a**) Fonell-Larcker criterion.

|     | AT | BI | EE | EM | FC | IM | PA | PC | PE | PP | PR | PS | SI |
|-----|-----|-----|-----|-----|-----|-----|-----|-----|-----|-----|-----|-----|-----|
| AT1 | 0.764 | 0.288 | 0.258 | 0.201 | 0.238 | 0.257 | 0.134 | 0.342 | 0.315 | 0.185 | 0.313 | 0.265 | 0.216 |
| AT2 | 0.904 | 0.280 | 0.326 | 0.339 | 0.179 | 0.343 | 0.167 | 0.336 | 0.382 | 0.171 | 0.358 | 0.269 | 0.236 |
| AT3 | 0.928 | 0.333 | 0.440 | 0.481 | 0.200 | 0.458 | 0.195 | 0.414 | 0.419 | 0.250 | 0.418 | 0.438 | 0.173 |
| AT4 | 0.843 | 0.226 | 0.221 | 0.115 | 0.167 | 0.219 | 0.427 | 0.204 | 0.213 | 0.139 | 0.209 | 0.137 | 0.146 |
| BI1 | 0.294 | 0.885 | 0.360 | 0.444 | 0.380 | 0.405 | 0.311 | 0.499 | 0.418 | 0.468 | 0.412 | 0.462 | 0.426 |
| BI2 | 0.244 | 0.831 | 0.341 | 0.280 | 0.053 | 0.334 | 0.190 | 0.021 | 0.391 | 0.083 | 0.046 | 0.141 | 0.091 |
| BI3 | 0.348 | 0.942 | 0.440 | 0.372 | 0.299 | 0.449 | 0.349 | 0.488 | 0.426 | 0.323 | 0.515 | 0.410 | 0.295 |
| EE1 | 0.220 | 0.265 | 0.785 | 0.255 | 0.163 | 0.377 | 0.169 | 0.501 | 0.445 | 0.285 | 0.493 | 0.434 | 0.211 |
| EE2 | 0.282 | 0.346 | 0.854 | 0.287 | 0.169 | 0.376 | 0.186 | 0.548 | 0.464 | 0.323 | 0.537 | 0.474 | 0.285 |
| EE3 | 0.289 | 0.393 | 0.840 | 0.244 | 0.213 | 0.369 | 0.174 | 0.549 | 0.417 | 0.280 | 0.494 | 0.425 | 0.333 |
| EE4 | 0.476 | 0.428 | 0.841 | 0.201 | 0.304 | 0.320 | 0.251 | 0.560 | 0.562 | 0.338 | 0.515 | 0.583 | 0.449 |
| EM1 | 0.452 | 0.455 | 0.432 | 0.900 | 0.378 | 0.563 | 0.287 | 0.480 | 0.383 | 0.465 | 0.546 | 0.588 | 0.494 |
| EM2 | 0.227 | 0.480 | 0.344 | 0.864 | 0.293 | 0.572 | 0.278 | 0.421 | 0.473 | 0.397 | 0.454 | 0.569 | 0.331 |
| FC1 | 0.192 | 0.238 | 0.189 | 0.263 | 0.749 | 0.168 | 0.217 | 0.256 | 0.213 | 0.226 | 0.225 | 0.294 | 0.238 |
| FC2 | 0.198 | 0.329 | 0.193 | 0.313 | 0.826 | 0.240 | 0.254 | 0.258 | 0.266 | 0.153 | 0.264 | 0.336 | 0.216 |
| FC3 | 0.184 | 0.243 | 0.177 | 0.232 | 0.704 | 0.166 | 0.287 | 0.198 | 0.235 | 0.160 | 0.234 | 0.265 | 0.211 |
| FC4 | 0.177 | 0.308 | 0.270 | 0.364 | 0.833 | 0.280 | 0.331 | 0.351 | 0.335 | 0.268 | 0.338 | 0.411 | 0.259 |
| IM1 | 0.468 | 0.581 | 0.573 | 0.580 | 0.274 | 0.918 | 0.334 | 0.550 | 0.532 | 0.363 | 0.571 | 0.496 | 0.366 |
| IM2 | 0.296 | 0.622 | 0.430 | 0.595 | 0.241 | 0.908 | 0.345 | 0.438 | 0.410 | 0.289 | 0.480 | 0.368 | 0.232 |
| PA1 | 0.187 | 0.201 | 0.148 | 0.179 | 0.212 | 0.212 | 0.755 | 0.196 | 0.128 | 0.168 | 0.144 | 0.140 | 0.207 |
| PA2 | 0.206 | 0.261 | 0.166 | 0.203 | 0.227 | 0.260 | 0.814 | 0.206 | 0.161 | 0.097 | 0.168 | 0.157 | 0.198 |
| PA3 | 0.174 | 0.202 | 0.111 | 0.192 | 0.260 | 0.228 | 0.725 | 0.217 | 0.184 | 0.098 | 0.249 | 0.167 | 0.137 |
| PA4 | 0.209 | 0.371 | 0.261 | 0.334 | 0.322 | 0.365 | 0.721 | 0.452 | 0.261 | 0.326 | 0.391 | 0.335 | 0.218 |
| PC1 | 0.323 | 0.368 | 0.573 | 0.399 | 0.304 | 0.424 | 0.319 | 0.768 | 0.565 | 0.493 | 0.510 | 0.530 | 0.252 |
| PC2 | 0.387 | 0.474 | 0.507 | 0.439 | 0.325 | 0.451 | 0.363 | 0.861 | 0.596 | 0.555 | 0.513 | 0.580 | 0.328 |
| PC3 | 0.352 | 0.527 | 0.535 | 0.380 | 0.312 | 0.468 | 0.329 | 0.875 | 0.503 | 0.461 | 0.499 | 0.494 | 0.328 |
| PC4 | 0.260 | 0.317 | 0.351 | 0.425 | 0.160 | 0.384 | 0.212 | 0.873 | 0.362 | 0.388 | 0.471 | 0.345 | 0.213 |
| PE1 | 0.249 | 0.290 | 0.445 | 0.350 | 0.196 | 0.327 | 0.169 | 0.483 | 0.740 | 0.466 | 0.516 | 0.568 | 0.183 |
| PE2 | 0.345 | 0.397 | 0.482 | 0.396 | 0.239 | 0.391 | 0.214 | 0.539 | 0.835 | 0.546 | 0.494 | 0.625 | 0.257 |
| PE3 | 0.342 | 0.446 | 0.444 | 0.375 | 0.298 | 0.398 | 0.223 | 0.538 | 0.820 | 0.513 | 0.561 | 0.586 | 0.285 |
| PE4 | 0.403 | 0.478 | 0.522 | 0.398 | 0.339 | 0.523 | 0.230 | 0.544 | 0.874 | 0.512 | 0.523 | 0.527 | 0.419 |
| PP1 | 0.151 | 0.293 | 0.355 | 0.358 | 0.208 | 0.317 | 0.202 | 0.478 | 0.518 | 0.758 | 0.503 | 0.491 | 0.170 |
| PP2 | 0.173 | 0.360 | 0.317 | 0.421 | 0.236 | 0.269 | 0.246 | 0.483 | 0.504 | 0.850 | 0.504 | 0.567 | 0.262 |
| PP3 | 0.215 | 0.442 | 0.325 | 0.383 | 0.270 | 0.308 | 0.214 | 0.520 | 0.517 | 0.857 | 0.517 | 0.509 | 0.296 |
| PP4 | 0.203 | 0.186 | 0.153 | 0.351 | 0.086 | 0.213 | 0.108 | 0.359 | 0.357 | 0.714 | 0.299 | 0.297 | 0.154 |
| PR1 | 0.450 | 0.479 | 0.560 | 0.355 | 0.294 | 0.544 | 0.296 | 0.506 | 0.414 | 0.387 | 0.779 | 0.539 | 0.387 |
| PR2 | 0.268 | 0.329 | 0.493 | 0.345 | 0.224 | 0.408 | 0.221 | 0.401 | 0.537 | 0.464 | 0.767 | 0.551 | 0.161 |
| PR3 | 0.349 | 0.444 | 0.527 | 0.408 | 0.259 | 0.448 | 0.281 | 0.549 | 0.412 | 0.551 | 0.833 | 0.430 | 0.243 |
| PR4 | 0.363 | 0.502 | 0.493 | 0.374 | 0.312 | 0.446 | 0.275 | 0.459 | 0.581 | 0.523 | 0.810 | 0.590 | 0.293 |
| PS1 | 0.416 | 0.492 | 0.513 | 0.462 | 0.336 | 0.536 | 0.250 | 0.491 | 0.591 | 0.476 | 0.531 | 0.839 | 0.465 |
| PS2 | 0.164 | 0.325 | 0.451 | 0.455 | 0.284 | 0.262 | 0.152 | 0.467 | 0.562 | 0.469 | 0.573 | 0.747 | 0.245 |
| PS3 | 0.277 | 0.463 | 0.491 | 0.537 | 0.359 | 0.322 | 0.248 | 0.532 | 0.543 | 0.543 | 0.451 | 0.840 | 0.350 |
| PS4 | 0.275 | 0.531 | 0.473 | 0.534 | 0.420 | 0.328 | 0.276 | 0.531 | 0.628 | 0.518 | 0.537 | 0.834 | 0.368 |
| SI1 | 0.184 | 0.296 | 0.284 | 0.326 | 0.267 | 0.223 | 0.162 | 0.288 | 0.282 | 0.220 | 0.263 | 0.373 | 0.767 |
| SI2 | 0.279 | 0.307 | 0.370 | 0.498 | 0.233 | 0.339 | 0.220 | 0.303 | 0.370 | 0.205 | 0.339 | 0.400 | 0.872 |
| SI3 | -0.041 | 0.224 | 0.164 | 0.186 | 0.103 | 0.148 | 0.101 | 0.135 | 0.127 | 0.178 | 0.119 | 0.217 | 0.717 |
| SI4 | 0.145 | 0.304 | 0.330 | 0.274 | 0.249 | 0.198 | 0.260 | 0.283 | 0.267 | 0.277 | 0.272 | 0.321 | 0.862 |

(**b**) Cross loading.

The size of the square root of the AVE of the endogenous IM factor was 0.913, being superior to the correlation scores displayed vertically (0.913 > 0.372, 0.542, 0.574, 0.358, 0.533, 0.475 and 0.330) and horizontally (0.913 > 0.421, 0.558, 0.555, 0.543, 0.282). The correlations of the constructs AT (0.236), BI (0.382), EE (0.406), EM (0.473), FC (0.296), IM (0.330), PA (0.257), PC (0.354), PE (0.380), PP (0.288), PR (0.358) and PS (0.458) also became lower in relation to the AVE square root of the Social Influence factor (0.736). The highest AVE values were obtained by the latent variables IM (0.913) and EM (0.883), and the lowest by SI (0.736) and BI (0.750), respectively.

Therefore, the construct model fulfills the condition of discriminated validity, since the variation shared with its own block of indicators of each factor (construct) exceeds the variation it shares with any other latent variable. The standardized average residual square root (SRMR) indicates the relevance of the proposed model. If the difference between the observed correlation matrix and the implicit correlation matrix is less than 0.08 [112] or reaches a maximum of 0.10 [113] then the relevance is proven. The proposed model is good and relevant as the mean size of this difference (SRMR) was 0.072 (<0.80).

If the indicators load well on the intended variables, and the cross-loads with other latent variables that they are not intended to measure are low, then the reflective model fulfills the condition of discriminated validity. Previous studies indicate the existence of a simple factor structure in a constructed model provided that the expected loads are greater than 0.7 (some use 0.6) and the cross-loads are below 0.6 [114]. Table 4a shows the existence of simple structures of factors that strengthen the label significance of the other latent variables due to cross-loads below level 0.6. The BI factor displays substantial cross-loads with the IM indicator (IM1: 0.581 > 0.6; IM: 0.522 > 0.6) and optimal incarnations on its own items (BI1: 0.885 > 0.7; BI2: 0.831 > 0.7; BI3: 0.942 > 0.7). If each factor of the model indicates a lower correlation with any other latent variable than with itself, the model is valid. Thus, the loads of the endogenous IM factor on its own construction exceed the minimum acceptable threshold of 0.7 (IM1 (0.918); IM2 (0.908)); while significant cross-loads were identified by the indicators EM2 (0.572), EM1 (0563), PR1 (0544), PE4 (0.523) or less marked with SI3 (0.148), FC1 (0.169) and FC3 (0.166).

Similarly, the EM factor obtained high loads on EM1's own articles (0.900): EM1 (0.864) and high cross-charges with IM2 (0.595); IM1 (0.580); PS4 (0.534) (Table 4b). Therefore, the discriminated validity condition was again met using cross-load size as an alternative to AVE.

The discriminated validity of a suitable model can be easily detected using the HTMT report. It expresses the ratio between the geometric average of the correlations of the indicators from the constructs that have the role of measuring different phenomena and the average of the correlations of the articles from the same construction [115]. Various studies have shown that for reflective models the HTMT value should be below 1.0 [116] or more strictly, be lower than 0.9 [117] or even 0.85 [118]. The results demonstrated maximum HTMT values between PE and PR (0.793) and minimum between PA and EE factors (0.268), conforming to the value threshold of 0.85 [119].

Multicollinearity in OLS regression is problematic if the variance inflation coefficient (VIF) is greater than 4.0 (some studies use a milder limit of 5.0) [112,117]. The reflective construct model also indicates very good multicollinearity, as the variance inflation coefficients (VIF) for each indicator, calculated with the OLS regression method, are lower than 4.0 [105]. VIF highs were reached by indicators AT2 (3117) and AT3 (3175), and lows of BI2 (1011), PA4 (1147), AT4 (1170) and PP4 (1210) (Figure 7).

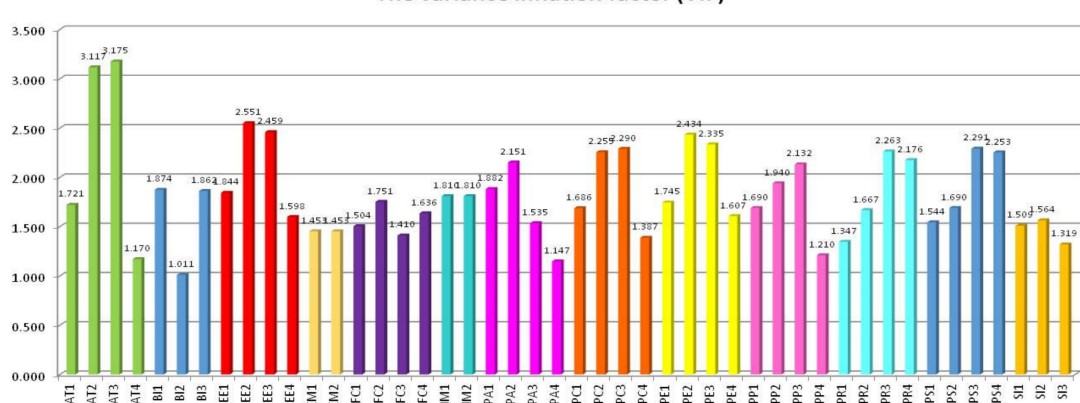

**Figure 7.** The variance inflation factor (VIF).

The GoF size of 0.595 indicates, after evaluating the data sets, a good level of fit of the proposed measurement model, being located almost in the middle of the allowed range (between 0 and 1) [120]

### 3.2. The Structural Model

The use of PLS algorithms allowed the evaluation of the structural relationships between the model constructs and the testing of research hypotheses. Table 5 shows the results of the correlations between the construct variables, values of the structural coefficients Path (β) and *t*-value correspondence (T-Statistics). *p*-values, $R^2$ and $Q^2$, after using the Bootstrapping and Blindfolding options (per 1000 copies) within SmartPLS.

**Table 5.** Results.

| Hypothesis | Correlation | β | T-Statistics | *p*-Values | Decision |
|------------|-------------|-------|--------------|------------|----------|
| $H_1$ | EE - > IM | 0.175 | 2.407 | 0.016 | Supported |
| $H_2$ | AT - > IM | 0.079 | 1.186 | 0.236 | No Supported |
| $H_3$ | PA - > IM | 0.123 | 2.484 | 0.013 | Supported |
| $H_4$ | PC - > IM | −0.010 | 0.110 | 0.912 | No Supported |
| $H_5$ | PR - > IM | 0.231 | 2.031 | 0.043 | Supported |
| $H_6$ | PE - > EM | 0.136 | 2.793 | 0.024 | Supported |
| $H_7$ | SI - > EM | 0.139 | 3.426 | 0.001 | Supported |
| $H_8$ | FC - > EM | 0.044 | 0.968 | 0.333 | No Supported |
| $H_9$ | PP - > EM | −0.004 | 0.057 | 0.954 | No Supported |
| $H_{10}$ | PS - > EM | 0.590 | 6.759 | 0.000 | Supported |
| $H_{11}$ | EM - > BI | 0.180 | 1.511 | 0.131 | No Supported |
| $H_{12}$ | EM - > IM | 0.344 | 4.149 | 0.000 | Supported |
| $H_{13}$ | IM - > BI | 0.542 | 5.879 | 0.000 | Supported |

The evaluation of the structural model included three distinct stages. In the first stage, the impact of exogenous variables was evaluated: Effort Expectancy, Attitude Towards IoT, Perceived Autonomy, Perceived Competence and Perceived Relatedness on the endogenous factor Intrinsic Motivation. Path coefficients (β) highlight the intensity of the links between the construct variables and their size must vary between 0 and 1 [121]. Loads of PR (0.231 > 0), EE (0.175) and PA (0.123) pathways show positive but low correlations in relation to IM (Table 5). A negative correlation was identified between the latent variables PC and IM (β = −0.010 < 0). The calculation results for the correlations EE- > IM (2407), PA- > IM (2484) and PR- > IM (2031) indicate t-values over 1.96, significant at the probability level of 0.05. The situation is reconfirmed by *p*-values which once again highlights that the three paths are significant at a level higher than the probability level 0.001. The AT- > IM and PC-IM

paths that showed *t*-values of 1.186 and 0.011 (< 1.96), respectively, are discordant, being insignificant at the level of 0.05.

The second stage aimed at determining the impact of the factors: Performance Expectancy (PE), Social Influence (SI), Facilitating Conditions (FC), Perceived Privacy (PP) and Perceived Security (PS) on Extrinsic Motivation (EM). In relation to the endogenous variable EM (Table 5) correlations were obtained: weak with the variables PP (−0.004 > 0) and FC (0.044), moderate with the factors SI (0.139), PE (0.136) but also high with PS - > MS (0.590 > 0). In this part of the construct the pathways PS - > EM (6.759 > 1.96), SI- > IM (3.426) and PR- > IM (2.301 > 1.96) proved to be positive and significant. Although insignificant, the PP -> EM (0.057) pathway indicates a negative impact compared to FC - > EM (0.968 < 1.96) which exerts an extremely weak but positive influence.

The third stage aimed at determining how intrinsic and extrinsic motivation influenced the behavioral intention to use IoT solutions by Romanian managers at work. Table 6 shows two strong correlations between the variables IM and BI (0.542) as well as between EM and IM (0.344) were. A positive but moderate correlation was identified between EM and BI (0.180 > 0). Of the three pathways linking endogenous variables, only two proved to be significant IM - > BI (5879 > 1.96) and EM - > IM (4149), respectively. Although the path between EM and BI (1,511) was positive, it remains insignificant as the *t*-value does not exceed the accepted threshold of 1.96 at a probability level of 0.05.

**Table 6.** The size of the coefficients of determination ($R^2$) and cross-redundancies ($Q^2$).

|      | $R^2$ | $R^2$ Adjusted | $Q^2$ |
| :---: | :---: | :---: | :---: |
| BI | 0.452 | 0.449 | 0.230 |
| EM | 0.630 | 0.625 | 0.462 |
| IM | 0.538 | 0.531 | 0.422 |

The coefficients of determination ($R^2$) indicate that 63% of the variation of the EM factor and 53.8% of the variation of the IM, respectively, is explained by the proposed construct model (Table 6). No $R^2$ quantities were generated for the rest of the elements, these being only exogenous latent factors [98,107]. As the adjusted $R^2$ is very close to the $R^2$ it follows that the predictors added to a regression model had a trivial correlation with the endogenous variable.

The endogenous factors BI (0.230), EM (0.462) and (0.422) modeled reflectively in this model obtained cross-redundancies (Stone–Gleisser value $Q^2$) over 0, indicating its relevance to predict these factors [101,122–124]. Figure 8a–c below shows the distribution of path load coefficients for the routes of the model from IM to BI, EM to BI and EM to IM, showing all information contained in confidence intervals.

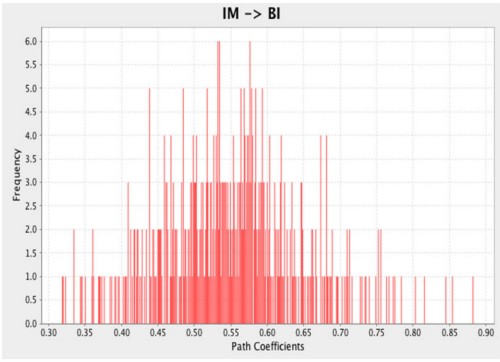

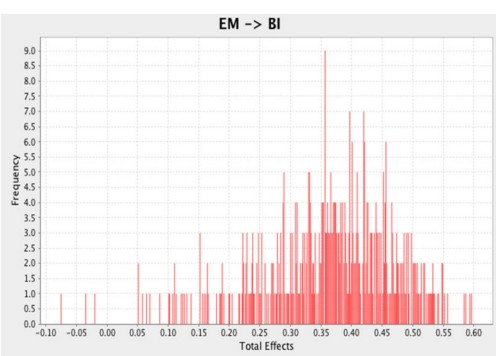

(**a**) Distribution of load coefficients for Intrinsic Motivation (IM) - > Behavioral Intention (BI)

(**b**) Distribution of load coefficients for Extrinsic Motivation (EM) - > BI

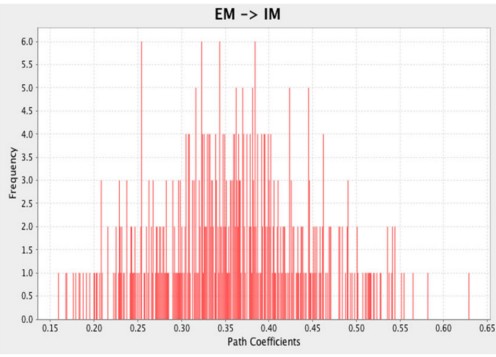

(**c**). Distribution of load coefficients for EM - > BI.

**Figure 8.** Distribution of load coefficients.

## 4. Conclusions

A new construct model was presented and validated, fulfilling the purpose of marketing research—identifying and analyzing the impact of factors in the spectrum of intrinsic and extrinsic motivations on the behavior of using IoT solutions among company managers in Romania in the future. A significant contribution was made both by developing the UTAUT construct model and by fulfilling the three objectives of our research. Thus, in assessing the extent to which exogenous variables had an impact on intrinsic motivation, it was shown that three out of five factors showed a positive and significant influence (Effort Expectancy (EE), Perceived Autonomy (PA) and Perceived Relatedness (PR)). At the same time, the direct and significant impact of the factors: Performance Expectancy (PE), Social Influence (SI) and Perceived Security (PS) on extrinsic motivation was illustrated. The positive and significant impact of the endogenous variable Intrinsic Motivation (IM) on the behavioral intention to use IoT solutions by managers of Romanian companies at work has been clearly demonstrated, even if that of Extrinsic Motivation remains only a positive one.

Corroborating the obtained results, the structural model explains 45.2% of the variance of the behavioral intention of Romanian managers to use IoT solutions at their workplace. In this context, eight of the thirteen hypotheses formulated were confirmed by empirical reality ($H_1$, $H_3$, $H_5$, $H_6$, $H_7$, $H_{10}$, $H_{12}$, $H_{13}$). In the first stage, the exogenous variables EE ($\beta = 0.175$; $p = 0.016$) and PA ($\beta = 0.123$; $p = 0.013$) had a positive and significant impact on the endogenous IM factor, confirming the acceptance of hypotheses $H_1$ and $H_3$. The construct model indicates a high empirical correlation (0.778) between the ease of understanding/understanding how to monitor, control and use IoT solutions (EE4) and the feeling of closeness to colleagues and business partners (PR1). On the other hand, there is a moderate positive empirical correlation (0.488) between the feeling that the use of IoT solutions will allow them

to better coordinate workplace activities (PA2) and the exchange of ideas through the IoT partner community (FC1).

Another result of the analysis confirms the null hypothesis ($H_5$) according to which Perceived Relatedness (PR) has a significant but strong impact on Intrinsic Motivation ($\beta = 0.231$, $p = 0.043$). Another significant and strong correlation was identified between the feelings of being well listened to by colleagues and partners (PR1) and the feelings of involvement, pleasure and satisfaction (IM1) (0.732). Unlike the other factors, the variable PC ($\beta = -0.010$) does not exert any impact on MI, while the exogenous factor AT ($\beta = 0.079$, $1.186 < 1.96$) shows a positive but insignificant impact. Our finding on the impact of latent EE, PA and PR variables on MI has been fully supported by other previous research [125,126].

In the second stage, the impact of the latent variables PE ($\beta = 0.136$; $p = 0.024$) and SI ($\beta = 0.139$; $p = 0.001$) on the exogenous factor EM was significant, reason for which the null hypotheses $H_6$ and $H_7$ were accepted. Given the use of IoT solutions by managers, the variable (PE4) that supports the acquisition of new interpersonal skills in programming shows a strong correlation (0.870) to the contribution to personal success (EM1). A positive but moderate correlation (0.558) was found between the help provided by the community in finding the best IoT solutions (SI2) and the recognition of personal achievements (EM2). At the level of the entire researched population, the Performance Security factor (PS) has a significant, strong impact ($\beta = 0.590$; $p = 0.000$) on the endogenous variable Extrinsic Motivation (EM). The strongest correlation, given the use of IoT solutions by managers, was found between the risk of cyberattacks on personal data (PS1) and the sense of contribution to professional success (EM1). The exogenous variable FC showed a positive and insignificant impact ($0.960 < 1.96$) on Extrinsic Motivation. The second negative impact of the construct was displayed between the latent variables Perceived Privacy (PP) and Extrinsic Motivation (EM). The influence of PE, SI and PS factors has been highlighted by other previous studies [127,128].

In the last stage of the construct model, the null hypotheses $H_{12}$ and $H_{13}$ were confirmed. Thus, the endogenous variable Extrinsic Motivation (EM) has a positive and strong impact on Intrinsic Motivation (IM) ($\beta = 0.344$, $p = 0.000$). Given the use of IoT solutions by managers, two strong correlations were found, the first (0.718) between recognition of personal achievements (EM2) and easy mastery of work tasks (IM2) and the second (0.663) between the feeling of contribution to my success (EM1) and that of involvement, pleasure and satisfaction (IM1).

The endogenous Intrinsic Motivation (IM) factor has a significant and strong impact on the variable Behavioral Intention to Use IoT solutions (BI) ($\beta = 0.542$, $p = 5.879 > 1.96$). Two other strong correlations are identified between the prediction of using IoT solutions in the future (BI3) and (0.666) the feeling of involvement, pleasure and satisfaction (IM1) and (0.703) the ability to easily master any task (IM2).

Regarding the impact of Extrinsic Motivation (EM) on Behavioral Intention to Use IoT solutions (BI), the alternative hypothesis was accepted ($\beta = 0.180$, $p = 1.511 < 1.96$), this impact being positive but insignificant. The influence exerted by the three exogenous variables Extrinsic Motivation (EM), Intrinsic Motivation (IM) and Behavioral Intention to Use (BI) have been the subject of numerous previous studies [19,94].

The results of this study have important and very useful implications for managers in Romania and also for those in other countries, regardless of the sectors of activity where they work, experience or location, highlighting the factors that should be considered and their impact. on the intrinsic and extrinsic motivation in substantiating the decisions and strategies regarding the implementation and use of IoT solutions within one's own activity, in order to improve future personal and business performance. A significant implication is the stimulation of intrinsic motivation due to the intention to use IoT solutions in the future. It is highlighted as a source of energy that pushes to action, an engine of the entire activity of managers, both personally and professionally, which aims to develop clear and concrete objectives such as: increasing the ability to monitor, control and use IoT solutions, identify new challenges and interact with the IoT ecosystem, identify options and opportunities for developing

desired activities, reduce the number of decisions and achieve more efficient results, get involved in managing and designing your own IoT solutions, participate in specialized courses, closeness and involvement in new relationships with colleagues and business partners and much more.

Another significant implication that contributes to the support of extrinsic motivation is the need to recognize that the acceptance or intention to use IoT solutions in the next period, by managers, should be managed with objectives such as: increasing personal productivity, faster work tasks, facilitating the exchange of ideas in the IoT partner community, providing the necessary support for the use of IoT solutions; the use of strong passwords to increase the security of personal information and reduce the risks of information manipulation, protection through the use of encryption technologies and ensuring better confidentiality and security of personal and business information.

An in-depth presentation of the factors influencing intrinsic and extrinsic motivation as well as their impact on behavioral intention to use IoT solutions will allow researchers to design and develop new more or less elaborate construct models that include new latent variables and indicators, highlighting new behaviors following the acceptance and use of IoT solutions. The results presented are important, given that this study is exceptional, unique and cannot be compared with other studies in the motivational and behavioral spectrum conducted in other fields.

The marketing research carried out has several limitations. Among them, we mention: low response rate (34%) and the generation of self-selection among respondents specific to online surveys, the exclusion of the influence of demographic moderators (gender, age, education level, etc.). In the future, other exploratory studies could improve the UTAUT theory by introducing new, unexplored latent variables and by creating added value in the context of the smart technology revolution.

**Author Contributions:** Conceptualization, M.C.T. and S.C.; methodology, M.C.T.; validation, M.C.T. and D.I.T.; formal analysis, A.I.S., M.Ș.H. and L.F.S.; writing—Original draft preparation, M.C.T., S.C.; writing—Review and editing, S.C. All authors have read and agreed to the published version of the manuscript.

**Funding:** This research received no external funding.

**Conflicts of Interest:** The authors declare no conflict of interest.

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
