# Peer review of "Motivations for the Use of IoT Solutions by Company Managers in the Digital Age: A Romanian Case"

_applsci, doi:10.3390/app10196905_

Round 1

Reviewer 1 Report

The article deals with: Motivations for the Use of IoT Solutions by Company Managers in the Digital Age: A Romanian Case. The aim of the paper is clear. The paper need more detailed description of the current state of knowledge in the researech area. The Materials and Methods section is broadly and logically described. Figure 7 is unclear, more needs to be explained. It is necessary to explain why demographic moderators, such as gender, have not been taken into account. The analytical part is very well done. It is necessary more explain the benefits of research and compare it with similar business studies in other countries.The article is interesting and after mentioned modifications I recommend it for publication.

Author Response

Reviewer 1

Comments and Suggestions for Authors

The article deals with: Motivations for the Use of IoT Solutions by Company Managers in the Digital Age: A Romanian Case. The aim of the paper is clear.

The paper need more detailed description of the current state of knowledge in the research area.

Response: The current state of knowledge in the research area was added in the Introduction section:

”Although users enjoy multiple benefits (Ferri et al., 2020; Lu et al., 2019; Caldarelli et al., 2017; Gangwar et al., 2015; Lee et al., 2015; Fagan et al., 2008), the proliferation of new communication technologies has raised privacy and security issues due to the unprecedented amounts of information it can collect, analyze and store (Waltzman & Shen, 2015). Researchers have conducted studies on the possible implications of IoT services on users' lives, and as a result, understanding user acceptance becomes an essential condition in the daily expansion of IoT applications. As IoT development is still in its infancy, few studies have been devoted to understanding acceptance issues from end users and especially from the perspective of company managers. Some researchers have theoretically investigated IoT through the prism of users, governments, and companies (Haller et al., 2009; Peoples et al., 2013; Weber, 2010; Zhao et al., 2013), implementation, architecture, and design (Khan et al., 2012; Gubbi et al., 2013). Other researchers have focused in their studies on investigating the negative aspects of IoT such as identification, heterogeneity, addressing and interoperability (Haller et al., 2009) or lack of acceptance of user acceptance as an obstacle in IoT adoption and deployment (Kim & Kim, 2016)”.

The Materials and Methods section is broadly and logically described.

Figure 7 is unclear, more needs to be explained.

Response: More details about multicollinearity in the reflective model proposed for research have been explained and introduced in the article

In OLS regression, multicollinearity exists when two or more independent variables are highly intercorrelated. Multicollinearity in OLS regression inflates standard errors, makes significance tests of independent variables unreliable, and prevents the researcher from assessing the relative importance of one independent variable compared to another. A common rule of thumb is that problematic multicollinearity may exist when the variance inflation factor (VIF) coefficient is higher than 4.0 (some use the more lenient cutoff of 5.0). VIF is the inverse of the tolerance coefficient, for which multicollinearity is flagged when tolerance is less than 0.25 (some use the more lenient cutoff of 0.20). (Hu & Bentler, 1999; Teo et al., 2008).

Tabel A. Collinearity Statistics (VIF)

Indicators

The variance inflation factor (VIF)

AT1

1.721

AT2

3.117

AT3

3.175

AT4

1.170

BI1

1.874

BI2

1.011

BI3

1.862

EE1

1.844

EE2

2.551

EE3

2.459

EE4

1.598

EM1

1.453

EM2

1.453

FC1

1.504

FC2

1.751

FC3

1.410

FC4

1.636

IM1

1.810

IM2

1.810

PA1

1.882

PA2

2.151

PA3

1.535

PA4

1.147

PC1

1.686

PC2

2.255

PC3

2.290

PC4

1.387

PE1

1.745

PE2

2.434

PE3

2.335

PE4

1.607

PP1

1.690

PP2

1.940

PP3

2.132

PP4

1.210

PR1

1.347

PR2

1.667

PR3

2.263

PR4

2.176

PS1

1.544

PS2

1.690

PS3

2.291

PS4

2.253

SI1

1.509

SI2

1.564

SI3

1.319

SI4

1.282

In our study there is no problematic multicollinearity because all variance inflation coefficients (VIF) are less than 4.0.

The data in Table A (above) were plotted in Figure 7 . We have improved Figure 7.

Figura 7. The variance inflation factor (VIF)

Consequently, I introduced the text:

"Multicollinearity in OLS regression is problematic if the coefficient of variance inflation factor (VIF) is greater than 4.0 (some studies use a milder limit of 5.0)" (Hu & Bentler, 1999; Teo et al., 2008).

It is necessary to explain why demographic moderators, such as gender, have not been taken into account.

Response: The measures used in modeling, including SmartPLS modeling, are often validated scales composed of several items, as in an "authoritarian leadership" scale. Measuring variables such as gender, age, education level, etc. with two or more articles do not lead to improved reliability nor do they improve the performance of the construct model. For example, if the gender variable (group 1-women and group 2-men) had been included in the model, the authors would have had to perform a multigroup analysis and determine whether the model is the same or different between groups. This comparison of models could highlight that the relationship between intrinsic and extrinsic motivation and the behavioral intention to use IoT solutions would be stronger in women than in men (or using other moderators could highlight a stronger relationship in young people compared to the elderly or in those with extensive experience or from different regions of the country.) (Venkatesh and Morris, 2000). In the future paper, we aim to highlight how the factors: sex, age, experience and location moderate the key relationships of the theory of acceptance or behavioral intention to use IoT solutions based on the UTAUT model.

The analytical part is very well done. It is necessary more explain the benefits of research and compare it with similar business studies in other countries.

Response: The advantages of the research and comparisons with other studies were included.

"Most previous research on the influence of intrinsic and extrinsic motivation on behavioral intention to use has focused more on issues of autonomy, control, performance, and facilitation (Davis 1992; Fagan, 2008; Cerasoli at all, 2014; Legault, 2016). However, no study has analyzed the relationship between intrinsic and extrinsic motivation variables and the intention to use IoT solutions in the future by different types of managers. Intrinsic and extrinsic motivations that determine managers to engage in competition, to develop new working relationships, to get actively involved in work and to achieve new performances, correlated with maintaining the confidentiality and security of personal data .”

The article is interesting and after mentioned modifications I recommend it for publication.

Thank you.

Reviewer 2 Report

This article conducts impact verification concerning intrinsic motivation, extrinsic motivation, and behavioral intentions for accepting and using the IoT solutions by the company managers in Romania. This article provides interesting and practical research topics for the IoT applications and scalability in the future. However, some questions about research methods still need to be answered.

  1. The authors use TAM or UTAUT model as the theoretical basis for technological acceptance, and use questionnaires verify the acceptance of the IoT systems. If the test samples (users) come from different companies, organizations, and industries, then the IoT system or technology used by all users (managers) needs to be the same system. However, this article does not explain whether the systems used in the questionnaire data collected refer to the same IoT system. And, what are the system functions, solutions and product features of the IoT used by these different companies? If the IoT systems used by different companies are different, there will have doubts about the reliability and validity of the IoT system. Although, the authors used the intended variables, the cross-loads with other latent variables and the criterion of Fornell-Larcker to prove the validity and reliability.
  2. As the author said that the users of IoT solutions include employees and managers. The author sets the research sample as managers and does not consider the opinions of employees. What is the purpose of this setting? The objectivity of the verification of TAM or UTAUT depended on the end users of the system. This article does not explain why the questionnaire data come from company managers for evaluation of the use of the IoT solutions.
  3. This article uses the SmartPLS3 Professional to verify the measurement model and the structure model. The line 394 is missing the Cronbach’s alpha value of PA, and the Cronbach’s alpha value of PP needs to be corrected.

  4. The conclusion mentioned that the null hypotheses H12 and H13 were confirmed.

    In line 553, the Intrinsic Motivation (IM) factor has a significant and strong impact on the variable Behavioral Intention to Use IoT solutions (BI) (β = 0.542, 5.879> 1.96) , is the p-value significant here?

    In line 558, the Extrinsic Motivation (EM) on Behavioral Intention to Use IoT solutions (BI), the alternative hypothesis was accepted (β = 0.180, 1.511 <1.96), is the p-value significant here?

Author Response

Reviewer 2

  1. The authors use TAM or UTAUT model as the theoretical basis for technological acceptance, and use questionnaires verify the acceptance of the IoT systems. If the test samples (users) come from different companies, organizations, and industries, then the IoT system or technology used by all users (managers) needs to be the same system. However, this article does not explain whether the systems used in the questionnaire data collected refer to the same IoT system. And, what are the system functions, solutions and product features of the IoT used by these different companies? If the IoT systems used by different companies are different, there will have doubts about the reliability and validity of the IoT system. Although, the authors used the intended variables, the cross-loads with other latent variables and the criterion of Fornell-Larcker to prove the validity and reliability.

Response: The proposed constructed model is based on the established UTAUT model. The IoT system used is the same. No "system functions, solutions and features of IoT Solutions" were presented in order not to advertise for free.

  1. As the author said that the users of IoT solutions include employees and managers. The author sets the research sample as managers and does not consider the opinions of employees. What is the purpose of this setting? The objectivity of the verification of TAM or UTAUT depended on the end users of the system. This article does not explain why the questionnaire data come from company managers for evaluation of the use of the IoT solutions.

Response: The analyzed sample included people who occupy the following positions within the company: Top Level Management, Middle Level Management and Lower Level Management. ( Table 1. Sample structure).

  1. This article uses the SmartPLS3 Professional to verify the measurement model and the structure model. The line 394 is missing the Cronbach’s alpha value of PA, and the Cronbach’s alpha value of PP needs to be corrected.

Response: The errors have been corrected.

“Six latent factors: BI (0.741), EM (0.717), FC (0.787), PA (0.757), PP (0.771) and SI (0.727) were….”

  1. The conclusion mentioned that the null hypotheses H12 and H13 were confirmed.

In line 553, the Intrinsic Motivation (IM) factor has a significant and strong impact on the variable Behavioral Intention to Use IoT solutions (BI) (β = 0.542, 5.879> 1.96) , is the p-value significant here?

In line 558, the Extrinsic Motivation (EM) on Behavioral Intention to Use IoT solutions (BI), the alternative hypothesis was accepted (β = 0.180, 1.511 <1.96), is the p-value significant here?

Response: The errors have been corrected.

“(β = 0.542, p - value=5.879 > 1.96).”

“(β = 0.180, p- value = 1.511 <1.96),”

Reviewer 3 Report

Dear authors,

thank you for your paper. Here I list few comments that should be useful to improve the quality of your paper.

Abstract.

The abstract should be modified.

The phrase “The evolution of the business environment follows the technological evolution through the use of IoT solutions both by the employed staff and by the managers.” Is not useful plaese delete it. Please organize the abstract writing 1) purpose 2) design/methology 3) Findings 4) research implication 5) originality of the paper.

Introduction.

From the introduction section a lack of literature should arise. I did not read any lack of literature in this paper. Also the contribution to literature should be enlarged and enforced (expecially with reference to the first one). You just identify wich are the contributes but you say nothing about.

Materials and methods.

I feel really unconfortable with this section that is really difficult to read.

First raw of the section IoT was not launched by Kevin Ashton. It is better if you write that the most common definition (or the first definition) of IoT was introduced by Kevin Ashton…

Please split section 2 in two different new section. I suggest you:

2) IoT definition and literature

3) Literature reviw about IOT and UTAUT

Please change the Hypotesis in Italic.

Please create a little space between different hypothesis

H11 and H13 are really weak hypothesis. Please enforce them.

Methods

In the methods section authors state “analyzed the relationship between intrinsic and extrinsic motivation variables and the intention to use IoT solutions in the future by different types of managers”. That’s great! But in my opinion this statement should stay in the introduction because it helps to identify the lack of literature and explain how you fill it.

Please organize the method section as follows:

  • The questionnaire
  • Sample selection
  • Methodology and data analisys
  • Questionnaire validation and preliminary results (with crombach’s alpha, AVE, etc)

Results and discussion

Please add a section with results and discussion. In this section provide just one table with all the results (this should improve the readability of results). I found several figure with the distribution of load coefficient, VIF and AVE. I think these figures can be summarized in tables making the paper more easy to read.

Conclusion

This section should be enforced too. What is new, what is the contribution and what are the implication of your results is something that I can just imagine. In the abstract authors state “The implications of this study are multiple for both managers and researchers.” However I did not found in the conclusion the word implication or something similar. This is the main weakness of the paper.

Other comments

Please add the table 1 in the text after the sample selection (and not in the appendix).

There are different grammatical errors and (expecially in the hypotesis part) the paperi s really hard to read. In my opinion the paper need to be proofreaded.

Overall I think authors should improve the quality of the paper by adding these references:

Fagan, M. H., Neill, S., & Wooldridge, B. R. (2008). Exploring the intention to use computers: An empirical investigation of the role of intrinsic motivation, extrinsic motivation, and perceived ease of use. Journal of Computer Information Systems48(3), 31-37.

Caldarelli, A., Ferri, L., & Maffei, M. (2017). Expected benefits and perceived risks of cloud computing: an investigation within an Italian setting. Technology Analysis & Strategic Management29(2), 167-180.

Gangwar, H., Date, H., & Ramaswamy, R. (2015). Understanding determinants of cloud computing adoption using an integrated TAM-TOE model. Journal of enterprise information management.

Ferri, L., Spanò, R., & Tomo, A. (2020). Cloud computing in high tech startups: evidence from a case study. Technology Analysis & Strategic Management32(2), 146-157.

Lee, Y., Lee, J., & Hwang, Y. (2015). Relating motivation to information and communication technology acceptance: Self-determination theory perspective. Computers in Human Behavior51, 418-428.

Lu, Y., Papagiannidis, S., & Alamanos, E. (2019). Exploring the emotional antecedents and outcomes of technology acceptance. Computers in Human Behavior90, 153-169.

Good Luck with your paper.

All my best.

Author Response

Reviewer 3

Comments and Suggestions for Authors

Dear authors,

thank you for your paper. Here I list few comments that should be useful to improve the quality of your paper.

Abstract.

The abstract should be modified.

The phrase “The evolution of the business environment follows the technological evolution through the use of IoT solutions both by the employed staff and by the managers.” Is not useful plaese delete it. Please organize the abstract writing 1) purpose 2) design/methology 3) Findings 4) research implication 5) originality of the paper.

Response: The phrase “The evolution of the business environment follows the technological evolution through the use of IoT solutions both by the employed staff and by the managers” was deleted.

The abstract was modified according to the reviewer’s request and at the end was introduced the following phrase indicating the originality of the paper: ” The originality of this article lies in the empirical part of the research, which, by using a quantitative method based on the questionnaire, provides important information on the impact of the motivational spectrum on the acceptance and use of IoT solutions by managers in the next period to achieve new performance, personal and professional”.

Introductions

From the introduction section a lack of literature should arise. I did not read any lack of literature in this paper. Also the contribution to literature should be enlarged and enforced (expecially with reference to the first one). You just identify wich are the contributes but you say nothing about.

Response: The following paragraph was added in the introduction section:

”Although users enjoy multiple benefits (Ferri et al., 2020; Lu et al., 2019; Caldarelli et al., 2017; Gangwar et al., 2015; Lee et al., 2015; Fagan et al., 2008), the proliferation of new communication technologies has raised privacy and security issues due to the unprecedented amounts of information it can collect, analyze and store (Waltzman & Shen, 2015). Researchers have conducted studies on the possible implications of IoT services on users' lives, and as a result, understanding user acceptance becomes an essential condition in the daily expansion of IoT applications. As IoT development is still in its infancy, few studies have been devoted to understanding acceptance issues from end users and especially from the perspective of company managers. Some researchers have theoretically investigated IoT through the prism of users, governments, and companies (Haller et al., 2009; Peoples et al., 2013; Weber, 2010; Zhao et al., 2013), implementation, architecture, and design (Khan et al., 2012; Gubbi et al., 2013). Other researchers have focused in their studies on investigating the negative aspects of IoT such as identification, heterogeneity, addressing and interoperability (Haller et al., 2009) or lack of acceptance of user acceptance as an obstacle in IoT adoption and deployment (Kim & Kim, 2016)”.

Materials and methods.

I feel really unconfortable with this section that is really difficult to read.

First raw of the section IoT was not launched by Kevin Ashton. It is better if you write that the most common definition (or the first definition) of IoT was introduced by Kevin Ashton…

Response: The word ”launched” was replaced with ”introduced” according to the reviewer’s request.

Please split section 2 in two different new section. I suggest you:

2) IoT definition and literature

3) Literature reviw about IOT and UTAUT

Response: Subsection 2.1. Was split in two according to the reviewer’s request:

2.1.1. IoT. Definition and evolution

2.1.2. UTAUT. Importance, advantages and limits

The old numbers were renumbered (2.1.3. and 2.1.4.).

Please change the Hypothesis in Italic.Please create a little space between different hypotheses.

H11 and H13 are really weak hypothesis. Please enforce them.

Response: Hypotheses were turned to Italic and a little space was created between different hypotheses, as reviewer’s request.

Methods

In the methods section authors state “analyzed the relationship between intrinsic and extrinsic motivation variables and the intention to use IoT solutions in the future by different types of managers”. That’s great! But in my opinion this statement should stay in the introduction because it helps to identify the lack of literature and explain how you fill it.

Please organize the method section as follows:

  • The questionnaire
  • Sample selection
  • Methodology and data analisys
  • Questionnaire validation and preliminary results (with crombach’s alpha, AVE, etc)

Response: The methods section has been reorganized according to the requirements of the reviewer.

2.2.1. The questionnaire

“The questionnaire included a formalized set of 54 questions (47 content questions and 7 identification questions), being specially designed to generate a new multidimensional model starting from the UTAUT elements [36]. Figure 1 shows the articles for the 13 latent variables included in the construct model. Effort Expectancy (EE) articles were adopted. Attitude Towards IoT (AT), Perceived Autonomy (PA), from the studies of authors Chen (2013), Economides (2017) and Wong (2002), while various new articles were introduced for Perceived Competence (PC) and Perceived Relatedness ( PR) from Ryan (2000) and Karahanna (1999) respectively. For Performance Expectancy (PE), Social Influence (SI), Facilitating conditions (FC) were adapted from the research of Davis (2003), Venkatesh (2012) and Economides (2017). Other new measurement elements were introduced for Perceived Privacy (PP) and Perceived Security (PS) from material from Albalawi (2018) and Dong (2014). The items that measure Intrinsic Motivation (IM) and Extrinsic Motivation have been adapted for managers who intend to use IoT solutions in the future. Gender, age, education level, management level, work experience, industry and location were the moderators used in our study. Respondents were asked about their work experience, managerial position and plans or intentions to use IoT solutions in the future. The use of a self-administered online questionnaire required the construction of short, closed questions that were subsequently measured using the 5-level Likert scale, from 1-strongly disagree to 5-strongly agree.”

2.2.4. Questionnaire validation and preliminary results

“Because the success of any marketing research depends overwhelmingly on the quality of the questionnaire in its design, two criteria were taken into account: ensuring the management of the information necessary to substantiate the decisions and the characteristics of the respondents. The model obtained from the design process was analyzed by a team of specialists who have solid knowledge, are involved in IoT projects and support services. After eliminating the double, confusing or unnecessary questions, its pretesting was performed in all aspects, on a restricted sample of respondents, the best data operators and respecting the proposed methodology. The results of the pre-testing of the questionnaire were analyzed very carefully, bringing the necessary improvements. After collection, the data were entered in the SPSS application to verify the correctness of the information provided by the respondents but also to eliminate the errors. Interactive data validation involves validating the questionnaire as soon as corrections have been made. Preliminary results (215 respondents in the previous period) for the latent variables Attitude Towards IoT (CA = 0.839, AVE = 0.682), Perceived Autonomy (0.752, AVE = 0.673, Perceived Privacy (CA = 0.746, AVE = 0.639), Social Influence (CA = 0.773, AVE = 0.697), Intrinsic Motivation (CA = 0.783, AVE = 0.814), indicated measures Cronbach’s Alpha (CA) greater than 0.6 [103] and Average Variance Extracted (AVE) greater than 0.5 [98].”

Results and discussion

Please add a section with results and discussion. In this section provide just one table with all the results (this should improve the readability of results).

Response: In the “Results and Discussions” section, to improve readability, all the results regarding the correlations between the construct variables were concentrated, in Table 4.

Tabel 4. Results

Hypothesis

Correlation

b

T Statistics

P Values

Decision

H1

EE -> IM

0.175

2.407

0.016

Supported

H2

AT -> IM

0.079

1.186

0.236

No Supported

H3

PA -> IM

0.123

2.484

0.013

Supported

H4

PC -> IM

-0.010

0.110

0.912

No Supported

H5

PR -> IM

0.231

2.031

0.043

Supported

H6

PE -> EM

0.136

2.793

0.024

Supported

H7

SI -> EM

0.139

3.426

0.001

Supported

H8

FC -> EM

0.044

0.968

0.333

No Supported

H9

PP -> EM

-0.004

0.057

0.954

No Supported

H10

PS -> EM

0.590

6.759

0.000

Supported

H11

EM -> BI

0.180

1.511

0.131

No Supported

H12

EM -> IM

0.344

4.149

0.000

Supported

H13

IM -> BI

0.542

5.879

0.000

Supported

I found several figure with the distribution of load coefficient, VIF and AVE. I think these figures can be summarized in tables making the paper more easy to read.

Response: A summary of the coefficients of variance inflation factor (VIF) for the construct model is presented in Table A (below).

Tabel A. Collinearity Statistics (VIF)

Indicators

The variance inflation factor (VIF)

AT1

1.721

AT2

3.117

AT3

3.175

AT4

1.170

BI1

1.874

BI2

1.011

BI3

1.862

EE1

1.844

EE2

2.551

EE3

2.459

EE4

1.598

EM1

1.453

EM2

1.453

FC1

1.504

FC2

1.751

FC3

1.410

FC4

1.636

IM1

1.810

IM2

1.810

PA1

1.882

PA2

2.151

PA3

1.535

PA4

1.147

PC1

1.686

PC2

2.255

PC3

2.290

PC4

1.387

PE1

1.745

PE2

2.434

PE3

2.335

PE4

1.607

PP1

1.690

PP2

1.940

PP3

2.132

PP4

1.210

PR1

1.347

PR2

1.667

PR3

2.263

PR4

2.176

PS1

1.544

PS2

1.690

PS3

2.291

PS4

2.253

SI1

1.509

SI2

1.564

SI3

1.319

SI4

1.282

In our study there is no problematic multicollinearity because all variance inflation coefficients (VIF) are less than 4.0.

The data in Table A (above) were briefly represented in Figure 7 . We have improved Figure 7. As can be seen, the presentation of the data in the table would be more extensive.

Figura 7. The variance inflation factor (VIF)

Response: Regarding the Average Variance Extracted (AVE) we entered the information in table 2.

Conclusion

This section should be enforced too. What is new, what is the contribution and what are the implication of your results is something that I can just imagine. In the abstract authors state “The implications of this study are multiple for both managers and researchers.” However I did not found in the conclusion the word implication or something similar. This is the main weakness of the paper.

Response: The “Conclusions” section has been renamed.

The implications for managers and researchers were also included here.

„A new construct model was presented and validated, fulfilling the purpose of marketing research – identifying and analyzing the impact of factors in the spectrum of intrinsic and extrinsic motivations on the behavior of using IoT solutions among company managers in Romania in the future. A significant contribution was made both by developing the UTAUT construct model and by fulfilling the three objectives of our research. Thus, in assessing the extent to which exogenous variables had an impact on intrinsic motivation, it was shown that three out of five factors showed a positive and significant influence (Effort Expectancy (EE), Perceived Autonomy (PA) and Perceived Relatedness (PR)). At the same time, the direct and significant impact of the factors: Performance expectancy (PE), Social influence (SI) and Perceived Security (PS) on extrinsic motivation was illustrated. The positive and significant impact of the endogenous variable Intrinsic Motivation (IM) on the behavioral intention to use IoT solutions by managers of Romanian companies at work has been clearly demonstrated, even if that of Extrinsic Motivation remains only a positive one.”

and

„The results of this study have important and very useful implications for managers in Romania and also for those in other countries, regardless of the sectors of activity where they work, experience or location, highlighting the factors that should be considered and their impact. on the intrinsic and extrinsic motivation in substantiating the decisions and strategies regarding the implementation and use of IoT solutions within one's own activity, in order to improve future personal and business performance. A significant implication is the stimulation of intrinsic motivation due to the intention to use IoT solutions in the future. It is highlighted as a source of energy that pushes to action, an engine of the entire activity of managers, both personally and professionally, which aims to develop clear and concrete objectives such as: increasing the ability to monitor, control and use IoT solutions, identify new challenges and interact with the IoT ecosystem, identify options and opportunities for developing desired activities, reduce the number of decisions and achieve more efficient results, get involved in managing and designing your own IoT solutions, participate in specialized courses , closeness and involvement in new relationships with colleagues and business partners and much more.

Another significant implication that contributes to the support of extrinsic motivation is the need to recognize that the acceptance or intention to use IoT solutions in the next period, by managers, should be managed with objectives such as: increasing personal productivity, faster work tasks, facilitating the exchange of ideas in the IoT partner community, providing the necessary support for the use of IoT solutions; the use of strong passwords to increase the security of personal information and reduce the risks of information manipulation, protection through the use of encryption technologies and ensuring better confidentiality and security of personal and business information.

An in-depth presentation of the factors influencing intrinsic and extrinsic motivation as well as their impact on behavioral intent to use IoT solutions will allow researchers to design and develop new more or less elaborate construct models that include new latent variables and indicators, highlighting new behaviors following the acceptance and use of IoT solutions. The results presented are important, given that this study is exceptional, unique and cannot be compared with other studies in the motivational and behavioral spectrum conducted in other fields.

Other comments

Please add the table 1 in the text after the sample selection (and not in the appendix).

Response: Table 1 has been introduced in the content of the article

There are different grammatical errors and (especially in the hypothesis part) the papers really hard to read. In my opinion the paper need to be proofreader.

Response: Grammatical errors have been corrected.

Overall I think authors should improve the quality of the paper by adding these references:

Fagan, M. H., Neill, S., & Wooldridge, B. R. (2008). Exploring the intention to use computers: An empirical investigation of the role of intrinsic motivation, extrinsic motivation, and perceived ease of use. Journal of Computer Information Systems48(3), 31-37.

Caldarelli, A., Ferri, L., & Maffei, M. (2017). Expected benefits and perceived risks of cloud computing: an investigation within an Italian setting. Technology Analysis & Strategic Management29(2), 167-180.

Gangwar, H., Date, H., & Ramaswamy, R. (2015). Understanding determinants of cloud computing adoption using an integrated TAM-TOE model. Journal of enterprise information management.

Ferri, L., Spanò, R., & Tomo, A. (2020). Cloud computing in high tech startups: evidence from a case study. Technology Analysis & Strategic Management32(2), 146-157.

Lee, Y., Lee, J., & Hwang, Y. (2015). Relating motivation to information and communication technology acceptance: Self-determination theory perspective. Computers in Human Behavior51, 418-428.

Lu, Y., Papagiannidis, S., & Alamanos, E. (2019). Exploring the emotional antecedents and outcomes of technology acceptance. Computers in Human Behavior90, 153-169.

Response: These 6 references were added in the Introduction section and at the end at the paper.

Thank you.

Round 2

Reviewer 3 Report

I found the paper improved and I think it can be considered for the publication.